# Optogenetic stimulation of anterior insular cortex neurons in male rats reveals causal mechanisms underlying suppression of the default mode network by the salience network

Vinod Menon [1,2,3,7] ✉, Domenic Cerri [4,5,6,7], Byeongwook Lee[1,7], Rui Yuan[1], Sung-Ho Lee [4,5,6] & Yen-Yu Ian Shih[4,5,6] ✉

The salience network (SN) and default mode network (DMN) play a crucial role in cognitive function. The SN, anchored in the anterior insular cortex (AI), has been hypothesized to modulate DMN activity during stimulus-driven cognition. However, the causal neural mechanisms underlying changes in DMN activity and its functional connectivity with the SN are poorly understood. Here we combine feedforward optogenetic stimulation with fMRI and computational modeling to dissect the causal role of AI neurons in dynamic functional interactions between SN and DMN nodes in the male rat brain. Optogenetic stimulation of Chronos-expressing AI neurons suppressed DMN activity, and decreased AI-DMN and intra-DMN functional connectivity. Our findings demonstrate that feedforward optogenetic stimulation of AI neurons induces dynamic suppression and decoupling of the DMN and elucidates previously unknown features of rodent brain network organization. Our study advances foundational knowledge of causal mechanisms underlying dynamic cross-network interactions and brain network switching.

Dynamic interactions between the salience network (SN) and default mode network (DMN) play a critical role in human brain function and dysfunction[1,2]. First identified from human neuroimaging data, based on task-related profiles of activation and deactivation during behavior, the SN is activated during attentionally demanding tasks while the DMN is typically suppressed[3–6]. Additionally, these activity changes are accompanied by concurrent increases in SN intra-network connectivity and decreased connectivity between SN and DMN[2,4,7]. These

networks have since been implicated in a wide range of cognitive tasks[3,8]. Specifically, the SN is a paralimbic-limbic network that plays a crucial role in identifying the most salient external events for adaptively guiding attention and behavior and for integrating cognitive, homeostatic, motivational, and affective signals[2,7]. In contrast, the DMN is a large-scale network anchored in cortical midline structures which is essential for internally oriented cognitive processes such as daydreaming, memory retrieval, future planning, and for integrating

[1]Department of Psychiatry & Behavioral Sciences, Stanford University School of Medicine, Stanford, CA 94305, USA. [2]Department of Neurology & Neurological Sciences, Stanford University School of Medicine, Stanford, CA 94305, USA. [3]Wu Tsai Neuroscience Institute, Stanford University School of Medicine, Stanford, CA 94305, USA. [4]Center for Animal MRI, University of North Carolina at Chapel Hill, Chapel Hill, NC 27599, USA. [5]Biomedical Research Imaging Center, University of North Carolina at Chapel Hill, Chapel Hill, NC 27599, USA. [6]Department of Neurology, University of North Carolina at Chapel Hill, Chapel Hill, NC 27599, USA. [7]These authors contributed equally: Vinod Menon, Domenic Cerri, Byeongwook Lee. ✉e-mail: menon@stanford.edu; shihy@unc.edu

social information[9]. Dysfunctional interactions between the SN and DMN are prominent in a wide range of psychiatric and neurological disorders including autism, attention deficit hyperactivity disorder, substance use disorder, depression, schizophrenia, and obsessive-compulsive disorder[8,10–12]. Consequently, the characterization of the neural mechanisms underlying DMN–SN interactions has significant clinical and translational relevance.

Non-invasive in vivo fMRI has played an important role in our understanding of the macroscopic functional organization of the SN and DMN. Computational analyses of causal dynamics in human fMRI data have suggested that the anterior insular cortex (AI), a key node of the SN, exerts the strongest inhibitory causal outflow to the DMN[4]. However, whether the AI directly suppresses the DMN, and influences dynamic interactions between the SN and DMN has not been empirically validated in vivo due to inherent constraints of human imaging, such as the unavailability of tools for selectively manipulating neuronal activity and network interactions. Furthermore, a lack of integration across scales and model systems has severely limited our understanding of how mesoscale neural processes influence macroscopic functional brain networks. Thus, to address these critical knowledge gaps, we use animal models and take advantage of invasive optogenetic tools, which provide versatile control of brain-circuit function by selective manipulations[13–15].

Causal manipulation of brain circuits with simultaneous whole-brain recordings has the potential to advance our understanding of the functional organization of large-scale brain networks in ways that resting-state fMRI alone cannot. Notably, a number of contradictory and inconsistent findings have been reported in the literature with respect to the intrinsic functional organization of the rodent SN and DMN[16–28]. While analysis of functional connectivity using resting-state fMRI has identified a rodent DMN anchored in the retrosplenial cortex (RSC)[17,23], analogous to the human posterior cingulate cortex and extended posterior medial cortex, there is less agreement about the inclusion of medial prefrontal cortex (mPFC) DMN nodes, such as the cingulate cortex (Cg) and prelimbic cortex (PrL)[20,22–24,26,27]. Indeed, in a recent study characterizing the rodent SN, Cg and PrL were implicated as nodes of the SN in addition to their putative roles in the DMN[17]. Recent imaging studies have also paradoxically assigned individual

subdivisions of the RSC and medial prefrontal cortex to both the SN and DMN[17,29–32]. Furthermore, the RSC is one of the largest cortical regions in rodents, and there is growing evidence for functional heterogeneity along its anterior/posterior (A/P) axis[16]. It follows that to accurately identify and model DMN–SN functional interactions, it is necessary to directly probe the functional involvement of Cg, PrL, as well as multiple RSC subdivisions in these networks. An important step in that direction is to probe circuit dynamics directly by selective manipulation of individual brain regions.

Here, we leverage optogenetic-fMRI technology[13,14,28,33–42] and recent advances in computational modeling of brain-circuit dynamics[43] to characterize the causal role of the AI in dynamic functional interactions between putative nodes of the rat SN and DMN including the AI, Cg, PrL, and multiple subdivisions of the RSC. We selectively applied feedforward optogenetic stimulation to Chronos-expressing neurons in the AI of experimental rats (hereafter referred to as Chronos rats) and enhanced yellow fluorescent protein-expressing neurons in the AI of control rats (EYFP rats) (Fig. 1a). We then used a Bayesian Switching Dynamic Systems (BSDS) state-space algorithm[43] to determine stimulation-induced changes in dynamic functional brain circuits associated with the SN and DMN (Fig. 2a). BSDS implements a hidden Markov model and unsupervised learning algorithm for determining latent brain states and dynamic switching processes from observed time-series data[43]. BSDS belongs to a class of latent space-switching models[44,45] which go beyond traditional methods for recovering the structure of the non-stationary time-varying organization of neural circuits[43,46,47]. Notably, BSDS does not require arbitrary sliding windows nor does it impose temporal boundaries associated with predefined task conditions[43]. BSDS applies a hidden Markov model to latent dynamic processes resulting in a parsimonious model of generators underlying the observed data, which is in contrast to approaches that apply a hidden Markov model directly to the observed fMRI data[48]. We used BSDS to evaluate the temporal dynamic properties of brain states, including the probability of occurrence of individual brain states, state transition probabilities, as well as time-varying activation and functional connectivity induced by AI stimulation (Fig. 2b).

We hypothesized that feedforward optogenetic stimulation of the AI would produce detectable brain states corresponding to periods

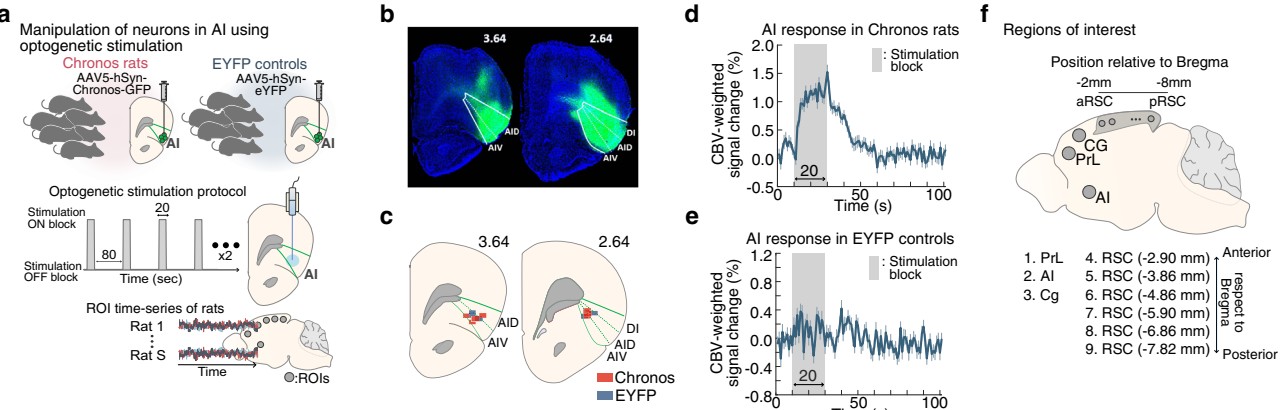

**Fig. 1 | Optogenetic stimulation of the anterior insular cortex (AI) with simultaneous fMRI in rats. a** We applied feedforward optogenetic stimulation to neurons of the right AI during fMRI in Chronos-expressing rats ($n = 9$) and EYFP-expressing controls ($n = 7$), and extracted time-series data from regions of interest (ROIs) corresponding to putative default mode network (DMN) and salience network (SN) nodes. **b** Representative histological confirmation of EYFP viral vector expression (green) in the dorsal agranular insular cortex (AID), ventral agranular insular cortex (AIV), and dysgranular insular cortex (DI) subdivisions of the right AI (white) in coronal brain slices at 2.64 mm and 3.64 mm anterior to Bregma. **c** Optical fiber placement was confirmed by anatomical MRI to be within the AID

and AIV subdivisions of the right AI (green) between 2.64 and 3.64 mm anterior with respect to Bregma in both Chronos and EYFP-expressing rats. **d** Averaged AI response relative to optogenetic AI stimulation blocks in Chronos rats. Data are presented as mean values ± SEM ($n = 9$ rats). **e** Averaged AI response relative to stimulation blocks in EYFP controls. Data are presented as mean values ± SEM ($n = 7$ rats). **f** Sagittal rat-brain cartoon highlighting ROIs used for fMRI time-series extraction, comprising the right AI, cingulate cortex (Cg), prelimbic cortex (PrL), and six subdivisions located at intervals of 1 mm along the anterior-posterior axis of the bilateral retrosplenial cortex (RSC), spanning from anterior RSC (aRSC) to posterior RSC (pRSC). Source data are provided as a Source Data file.

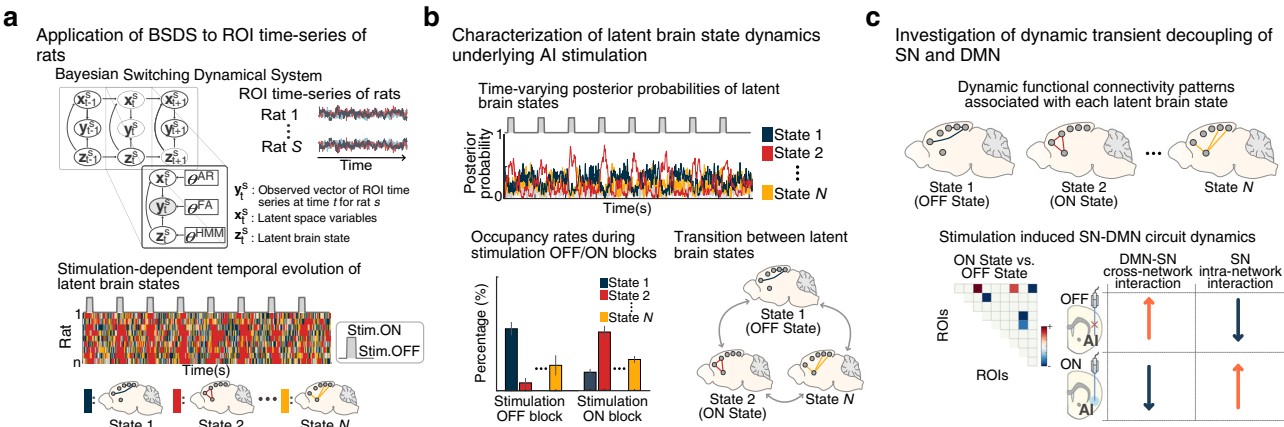

**Fig. 2 | Overall data analysis approach for identifying anterior insular cortex (AI)-stimulation-induced dynamic decoupling of the salience network (SN) and default mode network (DMN) in rats. a, b** We applied a novel Bayesian switching dynamical systems (BSDS) model to time-series data extracted from individual regions of interest (ROIs) corresponding to putative rat SN and DMN nodes to investigate brain state dynamics, and their temporal properties, associated with feedforward optogenetic stimulation of the AI. **c** We then examined dynamic changes in activation in each ROI and functional brain circuit connectivity associated with stimulation-related brain states.

with and without AI stimulation, each with a distinct pattern of activity and functional connectivity among putative SN and DMN nodes (Fig. 2c). We further predicted that AI stimulation would result in suppression of RSC activity (deactivation), decreased connectivity of the AI with the RSC, together with enhanced Cg and PrL activity and connectivity indicative of their differential roles in SN and DMN function and network dynamics. Based on prior findings of latent brain state dynamics during human cognition[43], we hypothesized that a state-space model-based approach would uncover dynamic brain circuits associated with SN and DMN interactions, including transient states which occurred between periods on and off AI stimulation. This allowed us to uncover brain state dynamics that were not time-locked to experimental task conditions and not detectable by conventional analyses.

In this work, our findings demonstrate that feedforward optogenetic AI stimulation induces dynamic suppression and decoupling of specific RSC subdivisions of the DMN, clarify causal mechanisms underlying dynamic cross-network interactions between the SN and DMN, and advance our understanding of rodent brain network organization.

## Results

### Optogenetic stimulation of AI and analysis of brain state dynamics

We applied feedforward optogenetic stimulation in an eight-epoch design to the right AI of nine Chronos rats and seven EYFP rats during cerebral blood volume (CBV) contrast-enhanced fMRI and extracted time-series data from nine regions of interest (ROIs) in the brain (Fig. 1a and Figs. S1–S3; see also Supplementary Information: Discussion #1 for the rationale underlying selective stimulation of the right AI). In each rat, the success of Chronos or EYFP viral vector expression was confirmed by histology (Fig. 1b; see also Supplementary Information: Note #1 for additional details about the anatomical profile of viral expression and optogenetic stimulation), and optical fiber placement was determined to be within the right AI by high-resolution anatomical T2-weighted MR images and individual locations were transcribed onto the Paxinos and Watson rat brain atlas, 6th edition[49] (Fig. 1c; see also Supplementary Information: Note #1 for additional details about the anatomical profile of viral expression and optogenetic stimulation), validating our manipulation locations. We confirmed the AI response to stimulation in the raw time-series data extracted from the AI ROI in ChR2 rats (Fig. 1d) and did not observe any qualitative changes to stimulation in the AI signal from EYFP controls (Fig. 1e).

We used BSDS to investigate dynamic changes in SN–DMN interactions associated with canonical nodes of the SN and DMN (see Supplementary Information: Methods #1 and 2 for BSDS model details; see also Supplementary Information: Discussion #2 for the key advantages of BSDS over other approaches). We chose anatomically-defined SN and DMN nodes in the AI, PrL, Cg, and RSC based on a wide range of published studies[16,17,20,22–27] (Fig. 1f and Table 1; see also Supplementary Information: Discussion #3 for the rationale for ROI selection). Notably, these regions have been the focus of investigations of the functional organization of the rodent SN and DMN. Based on prior reports of functional heterogeneity within the RSC[16,25,50], we divided this node into six subdivisions, spaced in 1 mm intervals along the anterior-to-posterior axis, allowing us to determine which subdivisions were most strongly coupled or decoupled with the AI during stimulation. Additionally, this facilitated the identification of heterogeneous responses and connectivity along the AP axis of the RSC. Our overall analysis strategy and procedures using BSDS are illustrated in Fig. 2. BSDS was applied

## Table 1 | Salience and default mode network nodes and their coordinates

| # | Region | Brain atlas coordinates | | |
|---|--------|------|------|------|
| | | **M–L** | **D–V** | **A–P** |
| 1 | Prelimbic cortex (PrL) | ±0.0 | −3.3 | +3.32 |
| 2 | The anterior insular cortex (AI) | +3.8 | −5.0 | +3.32 |
| 3 | Cingulate cortex (Cg) | ±0.6 | −2.4 | +1.32 |
| 4 | Retrosplenial cortex (RSC) | ±0.6 | −1.3 | −2.90 |
| 5 | | ±0.7 | −1.3 | −3.86 |
| 6 | | ±0.8 | −1.3 | −4.86 |
| 7 | | ±0.9 | −1.3 | −5.90 |
| 8 | | ±1.0 | −1.3 | −6.86 |
| 9 | | ±1.1 | −1.3 | −7.82 |

Coordinates are based on the Paxinos and Watson rat brain atlas (6th edition) coregistered to a group-template MR image space and represent the center of mass for each ROI. A–P, D–V, and M–L indicate millimeters from the skull at Bregma in anterior–posterior, dorsal–ventral, and medial–lateral directions, respectively. M–L coordinates were positive to the right and negative to the left of the midline. ± denotes the center of mass in each hemisphere for bilateral ROIs.

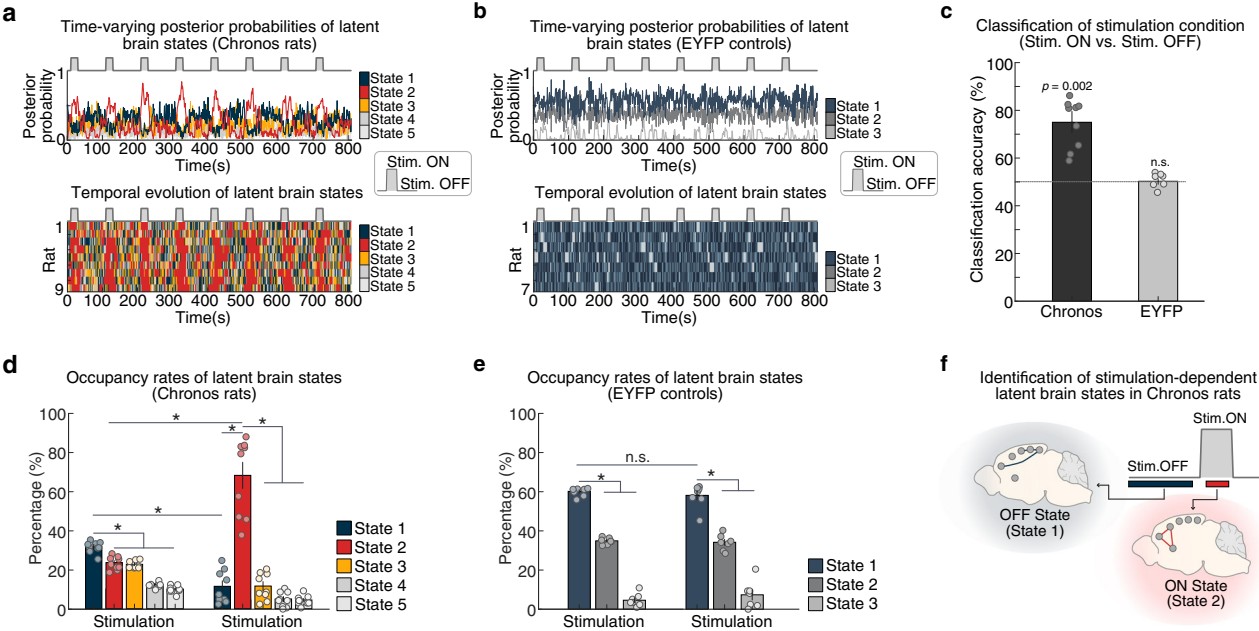

**Fig. 3 | Temporal properties of latent brain states associated with optogenetic stimulation of the anterior insular cortex (AI) in Chronos rats and EYFP controls. a** Time-varying posterior probabilities of brain states identified in Chronos rats by the Bayesian switching dynamical systems (BSDS) model across the AI stimulation protocol (top). Temporal evolution of the brain states identified in Chronos rats (bottom). **b** Time-varying posterior probabilities of the brain states in EYFP controls across the AI stimulation protocol (top). Temporal evolution of brain states identified in EYFP controls (bottom). **c** Classification analysis based on a linear Support Vector Machine classifier. Time-varying posterior probabilities of the latent brain states in Chronos rats ($n = 9$) distinguished stimulation ON and OFF blocks with a classification accuracy of 74.7% ($p = 0.002$, permutation test). In contrast, time-varying posterior probabilities of the latent brain states in EYFP controls ($n = 7$ rats) distinguished the stimulation condition only at the chance level (indicated by the black dashed line at 50%). **d** Occupancy rates of latent brain states in Chronos rats ($n = 9$). The occupancy rate of State 1 was significantly higher during stimulation OFF blocks compared to stimulation ON blocks. In contrast, the occupancy rate of State 2 was significantly higher during stimulation ON blocks compared to stimulation OFF blocks. **e** Occupancy rates of brain states in EYFP controls ($n = 7$ rats). State 1 dominated both stimulation ON and stimulation OFF blocks. **f** Schematic illustrating stimulation-dependent brain state dynamics in Chronos rats. Stimulation OFF blocks and stimulation ON blocks are dominated by State 1 and State 2, respectively. **c**–**e** Data are presented as mean values ± SEM. **d**, **e** *$p < 0.05$, n.s. $p \geq 0.05$, $p$-values determined by two-tailed $t$-test, with FDR correction; all exact $p$-values are provided in Source Data. Source data are provided as a Source Data file.

to ROI time-series data to identify brain states and determine their spatiotemporal properties associated with AI stimulation, including time-varying posterior probabilities and occupancy rates of each state, and transition probabilities between states (Fig. 2a, b). Finally, we examined the pattern of activation and functional connectivity among ROIs corresponding to each stimulation-related brain state in order to investigate dynamic functional interactions between putative nodes of the rat SN and DMN (Fig. 2c). Notably, BSDS allowed us to capture dynamic circuit properties that are missed by conventional approaches (see Figure S2 and Supplementary Information: Note #1 and Discussion #4 for details).

### Differentiation of brain states associated with optogenetic stimulation of AI in Chronos versus EYFP rats

Optogenetic stimulation of the AI in Chronos rats induced five distinct latent brain states defined by their unique spatiotemporal properties (see Materials and Methods for identification of latent brain states; also Fig. S4 and Supplementary Information: Note #3 for whole-brain, state-specific activation, and deactivation maps). In each rat, BSDS estimated the posterior probability of each latent brain state at each time point (Fig. 3a, top), and the latent brain state with the highest posterior probability was chosen as the dominant state at that point for that subject (Fig. 3a, bottom). Each latent brain state showed distinct moment-by-moment changes in posterior probability across time blocks with stimulation, and without stimulation, hereafter referred to as stimulation ON blocks and stimulation OFF blocks, respectively (Fig. 3a). In contrast, while three distinct latent brain states were

detected in EYFP rats, these did not show stimulation-dependent changes in posterior probability (Fig. 3b).

To further determine whether latent brain-state dynamics differentiate AI stimulation conditions, we conducted multivariate classification analyses using a Linear Support Vector Machine (LSVM) algorithm and leave-one-out cross-validation (LOOCV) (see Supplementary Information: Materials and Methods #3 for the stimulation block prediction). Classification models trained on posterior probabilities of latent brain states in Chronos rats accurately predicted stimulation ON blocks on unseen data with 74.7% accuracy, which significantly exceeds the chance level of 50% ($p = 0.002$, permutation test) (Fig. 3c). The classification analyses using the same approach on the posterior probabilities of latent brain states in the EYFP rats showed only a chance-level classification accuracy of 50% (Fig. 3c). These results demonstrate that the posterior probabilities of latent brain states distinguish stimulation ON from stimulation OFF blocks in Chronos, but not EYFP rats, thus providing evidence for specificity of brain states induced by AI stimulation.

### Temporal properties of latent brain state corresponding to stimulation ON and stimulation OFF blocks

We next examined occupancy rates of latent brain states during the stimulation ON and OFF blocks. Occupancy rate quantifies the fraction of time a given state is most likely to occur. Examination of AI stimulation effects on the temporal properties of each state revealed that State 1 has a significantly higher occupancy rate than other states during stimulation OFF blocks (all $ps < 0.05$, two-tailed $t$-test, FDR-

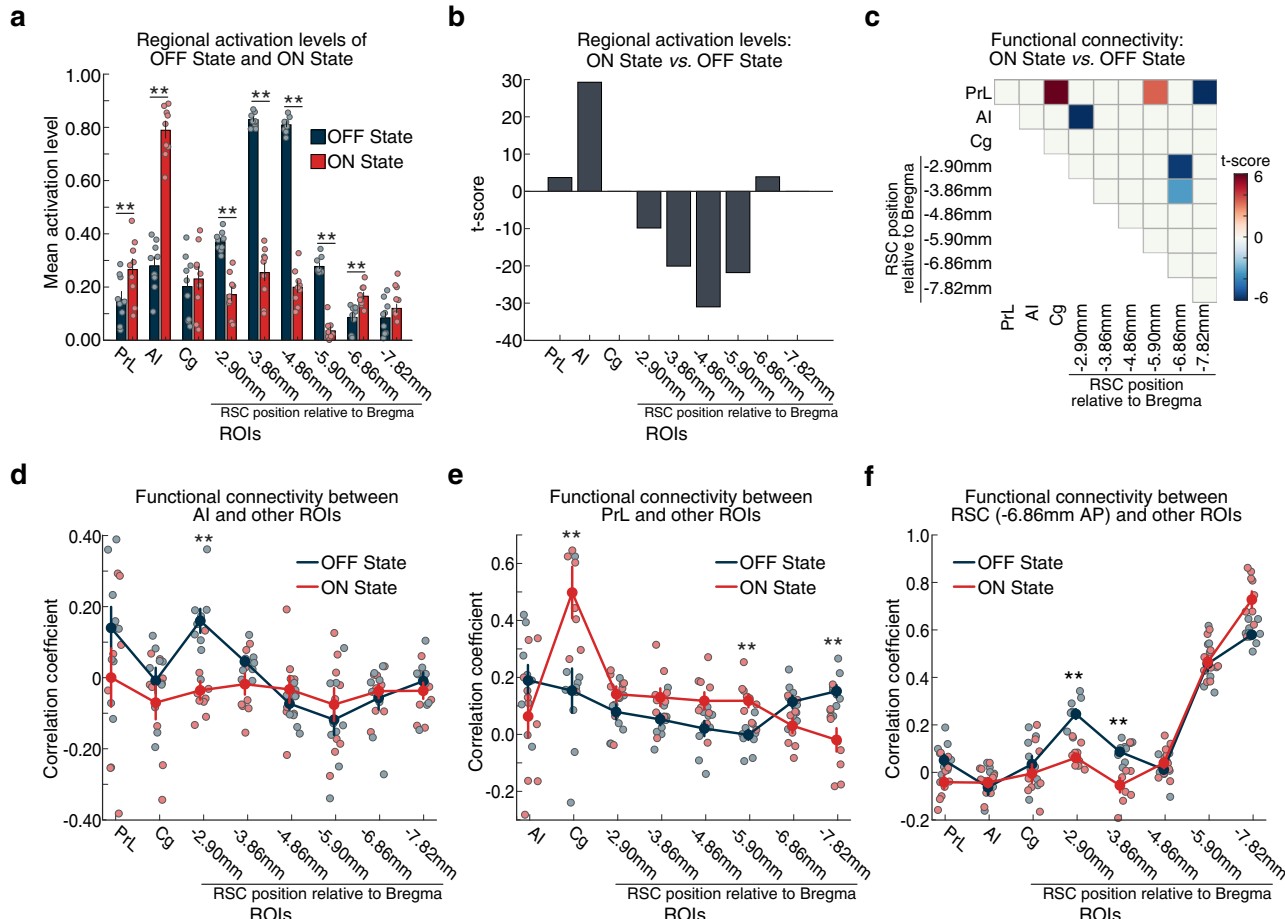

**Fig. 4 | Comparison of dynamic functional connectivity patterns in stimulation ON and OFF brain states in Chronos rats. a, b** Anterior insular cortex (AI) stimulation resulted in activation of AI and prelimbic cortex (PrL) and suppression of multiple subdivisions of the retrosplenial cortex (RSC) in the ON, compared to the OFF, state (*n* = 9 rats). **c** Specific links that showed significant differences in functional connectivity between the stimulation ON and OFF states (all *ps* < 0.01, two-tailed *t*-test, FDR-corrected, *n* = 9 rats; all exact *p*-values are provided in Source Data). Functional connectivity significantly decreased between the AI and an anterior subdivision of the RSC (−2.90 mm AP), within the RSC (between subdivisions at −2.90 and −3.86, and at −6.86 mm AP), and between the PrL and a mid-RSC subdivision (−5.90 mm AP). Furthermore, functional connectivity increased between the PrL and cingulate cortex (Cg), and the PrL and posterior RSC (−7.82 mm AP). **d–f** Functional connectivity changes between AI, PrL, and posterior RSC (−6.86 mm AP) with other ROIs during the ON and OFF states. **a, d–f** Data are presented as mean values ± SEM (*n* = 9 rats); **\**p* < 0.01, *p*-values determined by two-tailed *t*-test, with FDR correction; all exact *p*-values are provided in Source Data. Source data are provided as a Source data file.

corrected, Fig. 3d). Furthermore, the occupancy rate of State 1 was significantly higher during stimulation OFF compared to stimulation ON blocks (*p* < 0.05, two-tailed *t*-test, FDR-corrected, Fig. 3d), suggesting that State 1 is the dominant state associated with the stimulation OFF blocks (hence, the OFF state). In contrast, State 2 had a significantly higher occupancy rate than other states during stimulation ON blocks (all *ps* < 0.05, two-tailed *t*-test, FDR-corrected, Fig. 3d). Furthermore, the occupancy rate of State 2 was significantly higher during stimulation ON compared to stimulation OFF blocks (*p* < 0.05, two-tailed *t*-test, FDR-corrected, Fig. 3d), implying that State 2 is a dominant state associated with the stimulation ON blocks (hence, the ON state). In EYFP rats, a single state (i.e., State 1) dominated both stimulations ON and OFF blocks (all *ps* < 0.05, two-tailed *t*-test, FDR-corrected, Fig. 3e). Additional analyses using 10 s sub-blocks from the stimulation paradigm confirmed the predominance of States 1 and 2 across stimulation ON and OFF blocks, respectively (Fig. S5; see Supplementary Information: Note #4 for further details). Additional analyses revealed that the latent ON and OFF states have a mutually inhibitory influence on each other (Fig. S6; see Supplemental Information: Note #5 for further details). Taken together, these results

demonstrate that stimulation ON and OFF blocks are dominated by distinct latent brain states in Chronos rats, but not in EYFP rats (Fig. 3f).

## Differences in regional activation between the ON and OFF brain states

We examined the effects of AI stimulation on the activation and deactivation of each ROI during the ON and OFF brain states estimated by BSDS (Fig. 4a, b; see Supplementary Information: Materials and Methods #4 for details). BSDS model-based analysis identified strong evoked responses in the AI in Chronos rats (Fig. 4a). Notably, AI stimulation in Chronos rats resulted in suppression of multiple sections of the RSC spanning its anterior-to-middle subdivisions from −2.90 to −5.90 mm AP (Fig. 4a, b, all *ps* < 0.01, two-tailed *t*-test, FDR-corrected). In addition, this analysis revealed significant activation of the AI, PrL, and a posterior node of RSC at −6.86 mm AP (Fig. 4a, b, all *ps* < 0.01, two-tailed *t*-test, FDR-corrected), but no change in Cg activity. These results demonstrate the causal influence of AI stimulation on the suppression of the DMN, identify the RSC area from −2.90 to −5.90 mm AP as a critical DMN node, and point to the involvement of the AI and PrL in the SN.

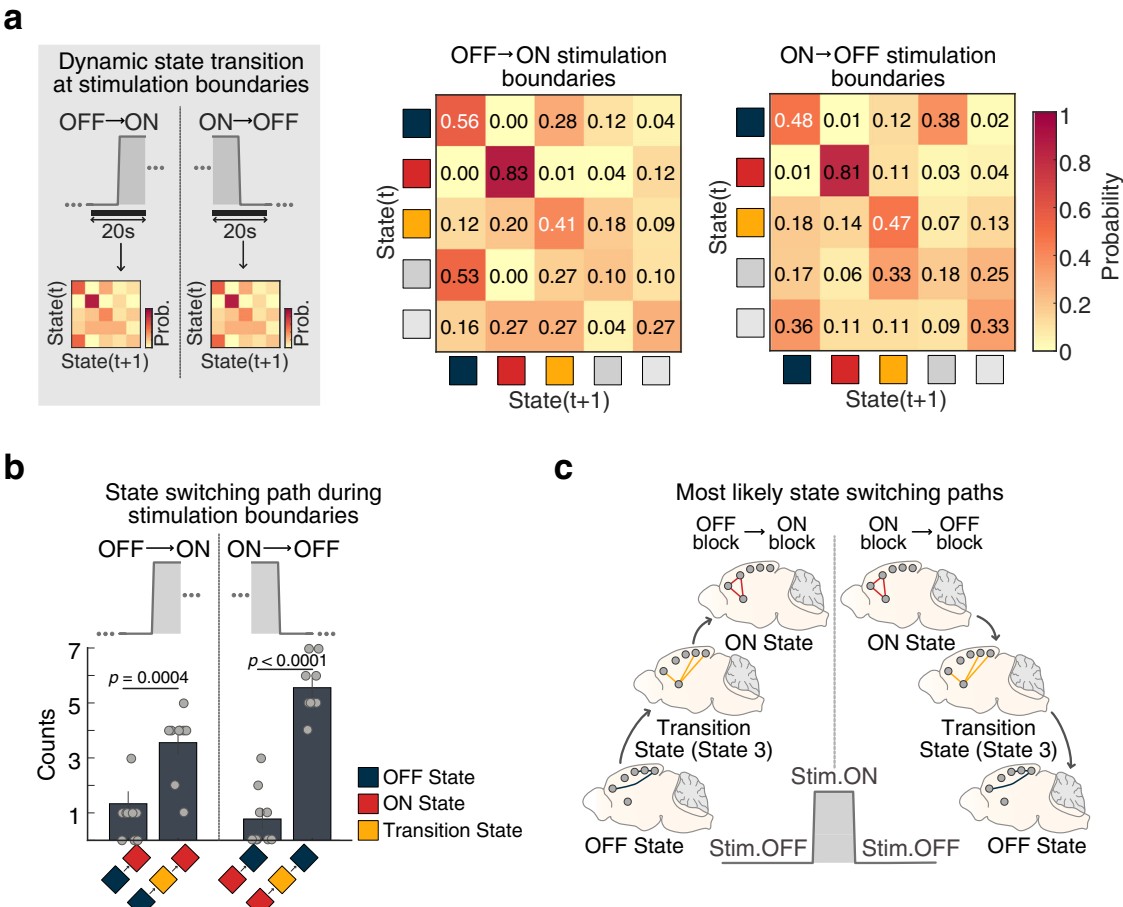

**Fig. 5 | Identification and spatiotemporal properties of a Transition state in Chronos rats. a** State switching probability of latent brain states at stimulation boundaries identified in Chronos rats. State switching probability is defined as the probability that a latent brain state at time instance ($t$) switches to other states at time instance ($t + 1$), or remains in the same state. At the OFF→ON stimulation boundary, the switching probability from the OFF to ON state (i.e., $P_{OFF \to ON}$) was significantly lower than the switching probabilities from the OFF to State 3 (i.e., $P_{OFF \to State3}$) and from State 3 to the ON state (i.e., $P_{State3 \to ON}$), respectively. Furthermore, at the ON→OFF stimulation boundary, the switching probability from the ON to OFF state (i.e., $P_{ON \to OFF}$) was significantly lower than the switching

probabilities from the ON state to State 3 (i.e., $P_{ON \to State3}$) and from the State 3 to OFF state (i.e., $P_{State3 \to OFF}$), respectively. All $ps < 0.001$, determined by two-tailed $t$-test, with FDR correction; all exact $p$-values are provided in Source Data; $n = 9$ rats. **b** Analysis of state switching matrices revealed that state transitions between the OFF state (State 1) and the ON state (State 2) first pass through a Transition state (State 3). Data are presented as mean values ± SEM ($n = 9$ rats); all $p$-values are determined by a two-tailed $t$-test. **c** Schematic illustrating the most likely state switching path from stimulation OFF to stimulation ON blocks. Source data are provided as a Source Data file.

## Differences in regional functional connectivity between the ON and OFF brain states

We then examined changes in dynamic functional connectivity associated with the ON and OFF brain states derived from the BSDS model in Chronos rats. Univariate link-specific analysis was conducted to determine unique functional connectivity patterns that differentiate the ON and OFF states (see Supplementary Information: Materials and Methods #4 for details). This analysis revealed that the ON state has unique connectivity patterns compared to the OFF state (all $ps < 0.01$, two-tailed $t$-test, FDR-corrected). Notably, the ON state showed decreased connectivity between AI and an anterior RSC region ($-2.90$ mm AP), suggesting decoupling between SN and DMN (Fig. 4c, d). In addition, the ON state also showed increased connectivity between the PrL and Cg, and the PrL and a middle RSC subdivision ($-5.90$ mm AP; Fig. 4c, e), but decreased connectivity between the PrL and posterior RSC ($-7.82$ mm AP; Fig. 4c, e) and between anterior and posterior RSC subdivisions (Fig. 4c, f). Direct estimation of functional connectivity using data from time points corresponding to the BSDS-derived ON and OFF states yielded convergent results and validated our findings (Figs. S7 and S8; see Supplementary Information: Note #6 for details). Collectively, these results suggest functional involvement

of the PrL and Cg and functional heterogeneity within the RSC underlying SN–DMN dynamics.

## Transient latent brain states induced by optogenetic stimulation in Chronos rats

BSDS also uncovered a third state (State 3, Fig. 3d) in Chronos rats. We used BSDS-derived state switching probabilities to investigate the functional role of this state at the boundary between the OFF and ON stimulation blocks (i.e., the Transition state, see Fig. S9 and Supplementary Information: Note #7 for details). Our analysis revealed that, at the OFF→ON stimulation boundary, the switching probabilities from the OFF to Transition state (i.e., $P_{OFF \to Transition}$) and from Transition to ON state (i.e., $P_{Transition \to ON}$) were both significantly higher than the switching probabilities from the OFF to ON state (i.e., $P_{OFF \to ON}$) (all $ps < 0.001$, two-tailed $t$-test, FDR-corrected, Fig. 5a). Similarly, at the ON→OFF stimulation boundary, the switching probabilities from the ON state to Transition state (i.e., $P_{ON \to Transition}$) and from the Transition state to OFF state (i.e., $P_{Transition \to OFF}$) were both significantly higher than the switching probability from the ON to OFF state (i.e., $P_{ON \to OFF}$) (all $ps < 0.001$, two-tailed $t$-test, FDR-corrected, Fig. 5a). Further analysis of counting switching paths between the OFF and ON states at the

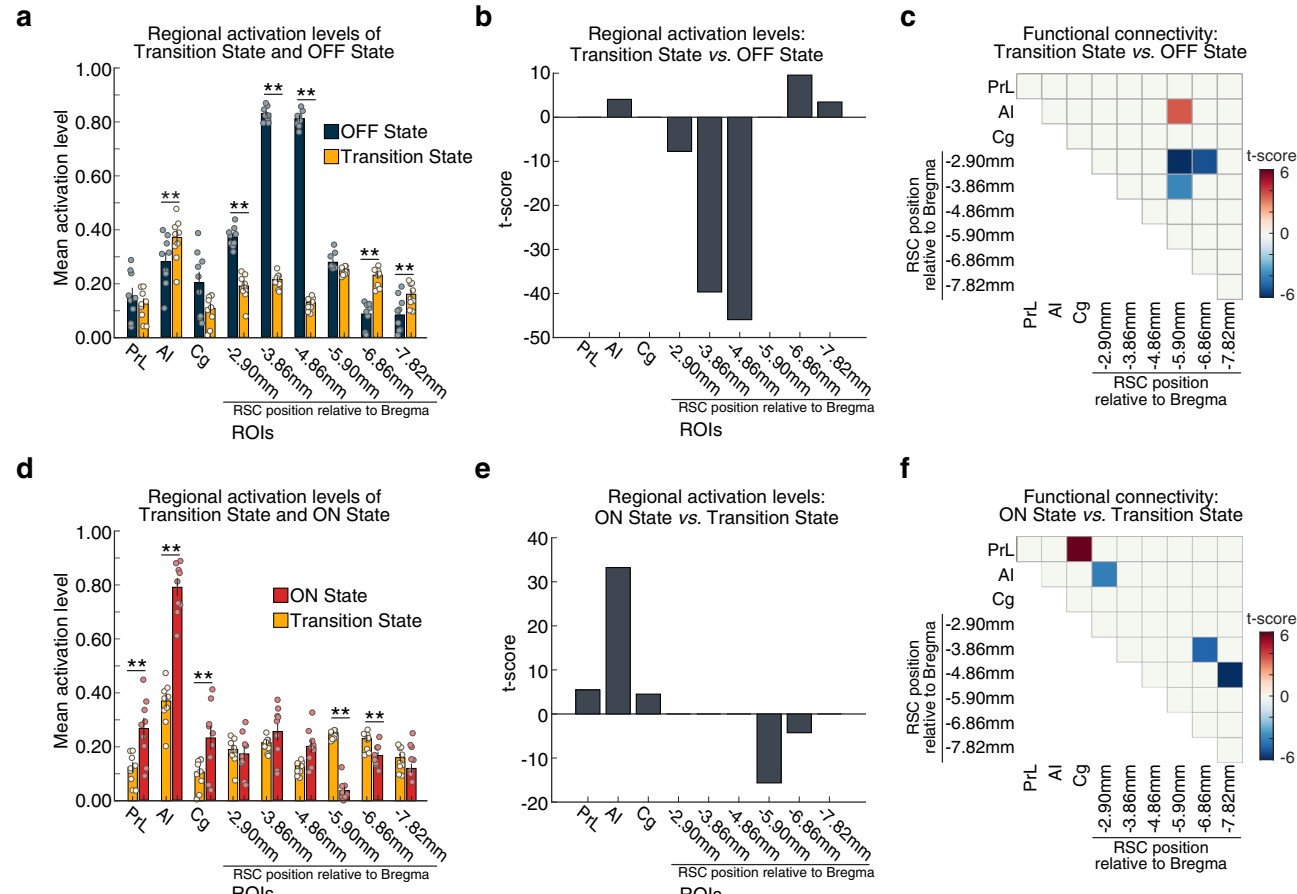

**Fig. 6 | Comparison of dynamic functional connectivity patterns of the Transition state with stimulation ON and OFF states in Chronos rats. a, b** Regional activation levels during Transition and OFF states (n = 9 rats). Anterior insular cortex (AI) stimulation resulted in a Transition state characterized by suppression of multiple anterior to the mid-region retrosplenial cortex (RSC) subdivisions, and activation of AI and multiple posterior RSC subdivisions. **c** Specific links that showed significant differences in functional connectivity between Transition and OFF states. Functional connectivity between AI and posterior RSC (−5.8 mm AP) significantly increased, while connectivity within RSC decreased between anterior and posterior subdivisions in the transition state, compared to the stimulation OFF state. **d, e** Regional activation levels during Transition and ON states (n = 9 rats). AI

stimulation resulted in activation of AI, prelimbic cortex (PrL), and cingulate cortex (Cg), and suppression of middle and posterior RSC (−5.8 and −6.8 mm AP) in the ON state, compared to the Transition state. **f** Specific links that showed significant differences in functional connectivity between the ON and Transition states. Functional connectivity significantly decreased between the AI and anterior RSC (−2.9 mm AP), and between anterior-middle and posterior subdivisions within the RSC. Furthermore, functional connectivity increased between the PrL and Cg. **a, d** Data are presented as mean values ± SEM (n = 9 rats); **p < 0.01, p-values determined by two-tailed t-test, with FDR-correction; all exact p-values are provided in Source Data. **c, f** All ps < 0.01, determined by two-tailed t-test, FDR-corrected; all exact p-values are provided in Source Data; n = 9 rats. Source data are provided as a Source Data file.

stimulation boundaries in each rat revealed that these states first pass through the Transition State rather than switching directly (all ps < 0.001, two-tailed t-test, Fig. 5b). Two other states (States 4 and 5) had low occurrence (Fig. 3d), and did not function as transition states between the OFF and ON states. Together, these results demonstrate that switching between OFF to ON states occurs via a robust Transition state (Fig. 5c).

### Comparison of activation and functional connectivity during the Transition state compared to the OFF state

Leveraging the power of latent brain models, we sought to determine changes in putative rat DMN and SN node activation and functional connectivity between the OFF and Transition brain states, as this is an intermediate step in the dynamic transition from the OFF to the ON state. Notably, in comparison to the OFF state, the Transition state following AI stimulation in Chronos rats showed suppression of multiple subdivisions of the RSC spanning its anterior to mid-regions (−2.90 to −4.86 mm AP), but activation of posterior subdivisions (−6.86 and −7.82 mm AP) (Fig. 6a, b, all ps < 0.01, two-tailed t-test, FDR-corrected). Further, while the AI was also activated (p < 0.01, two-tailed

t-test, FDR-corrected), Cg and PrL did not exhibit significant activity changes during the Transition state (Fig. 6a, b).

We next examined differences in functional connectivity between the Transition and OFF states. Our analysis revealed that, compared to the OFF state, the Transition state was characterized by stronger functional connectivity between the AI and a mid-region of the RSC (−5.90 mm AP), but reduced intra-RSC connectivity between anterior (−2.90 and −3.86 mm AP) and mid-posterior subdivisions of RSC (−5.90 and −6.86 mm AP) (Fig. 6c, all ps < 0.01, two-tailed t-test, FDR-corrected). There were no significant differences involving Cg and PrL nodes. Taken together, these results demonstrate that AI stimulation in rats results in transient brain states characterized by heterogeneous profiles of deactivation, activation, and connectivity along the anterior-posterior axis of the RSC.

### Comparison of activation and functional connectivity during the ON state compared to the Transition state

We then sought to determine how regional activation levels associated with AI stimulation change from the Transition state to the ON state. Our analysis revealed significant increases in AI, PrL, and Cg activation

during the ON state, compared to the Transition state. In contrast, the posterior subdivisions of RSC (−5.90 mm and −6.86 mm AP) were suppressed during the ON compared to the Transition state (Fig. 6d, e, all $ps < 0.01$, two-tailed $t$-test, FDR-corrected).

Analysis of differences in regional functional connectivity between the ON and Transition states revealed decreased connectivity in the ON state between the AI and an anterior RSC subdivision (−2.90 mm AP), suggesting decoupling between SN and DMN. In addition, the ON state also showed increased connectivity between the PrL and Cg but decreased connectivity between the anterior-middle RSC subdivisions (−3.86 mm, −4.86 mm AP) and posterior RSC (−6.86 mm, −7.82 mm AP) subdivisions (Fig. 6f, all $ps < 0.01$, two-tailed $t$-test, FDR-corrected). Overall, the results from the comparison of the ON state to the Transition state are in agreement with those from a comparison of the ON state to the OFF state. Direct estimation of functional connectivity using data from time points corresponding to the BSDS-derived Transition, ON, and OFF states yielded convergent results and validated our findings (Fig. S10). Collectively, these results provide further evidence for the functional involvement of the PrL and Cg nodes in SN-DMN dynamics, and suppression of RSC nodes of the DMN as well as functional heterogeneity within the RSC underlying SN–DMN interactions.

### Replication of findings using extended SN and DMN ROIs

To determine the robustness of our findings with respect to ROI selection, we conducted additional analyses by incorporating amygdala and hippocampus nodes, which are known to be part of SN[17] and DMN[23], respectively (Fig. S11). All major findings were replicated with this more general resting-state network-derived choice of ROIs (see Supplemental Information: Note #9 for details). All major findings were also replicated when striatal and medial temporal lobe (MTL) ROIs, uncovered by conventional general linear model analysis (Fig. S2), were included with the original SN and DMN nodes (see Fig. S12 and Supplemental Information: Note #10 for details). These findings suggest that the putative SN and DMN nodes originally selected to test our a priori hypotheses are representative of these networks, and provide replicable information to capture the latent brain states underlying dynamic SN–DMN interactions induced by optogenetic stimulation of the AI.

### Control analyses using ROIs located outside the SN and DMN

To further investigate the specificity of our findings with respect to ROI selection, we used BSDS to identify brain states from nodes in the auditory, visual, and motor cortex (Fig. S13). We found that none of the main findings reported above were observed with the latent brain state estimates from this analysis (see Supplementary Information: Note #11 for details). Taken together, these results demonstrate the specificity of our findings with respect to canonical SN–DMN ROIs.

## Discussion

We combined selective feedforward optogenetic manipulation with simultaneous fMRI to determine the causal role of the AI in a potentially antagonistic relationship between the SN and DMN in the rat brain. Our computational approach using state-space modeling revealed five key findings associated with AI feedforward stimulation: (1) Distinct stable brain states of patterned activity corresponding to periods with and without AI stimulation; (2) DMN suppression of specific subdivisions of the RSC; (3) Activation of the AI and PrL; (4) Decreased AI connectivity with RSC, increased PrL and Cg connectivity, and decreased connectivity between RSC subdivisions; and (5) A transition state corresponding to the periods between stimulation conditions, characterized by a heterogenous profile of RSC suppression and activation, as well as increased AI to RSC and decreased intra-RSC connectivity. These findings were specific to the stimulation

of Chronos-expressing neurons in experimental rats and were not observed in EYFP-expressing control rats.

Our study demonstrates that AI stimulation induces dynamic suppression of the DMN and decoupling of the DMN from the SN (see also Supplementary Information: Discussion #5 for details on the antagonistic functional relationship between the SN and DMN). In addition, PrL activation and enhanced PrL-Cg coupling during AI stimulation indicate that these regions, commonly included in the rodent DMN, also have a functional role in the SN (see also Supplementary Information: Discussion #6 for details about the heterogeneity of rodent PrL/mPFC network function)[17]. Furthermore, AI stimulation-induced heterogenous deactivation and connectivity patterns along the AP axis of RSC, which were particularly apparent during the brief transition state, point to the existence of distinct functional modules within the rodent RSC. Deep-brain structures have been implicated in DMN function and regulation[51,52] (see Supplementary Information: Discussion #7 for a summary of recent findings). Given that these structures are reciprocally connected with the AI[53,54], DMN-SN network dynamics in real-world situations are likely the product of deep-brain influences and the cortical interactions described here. Our findings reveal circuit mechanisms underlying suppression of the DMN and demonstrate a causal role for AI in network switching. Our findings also identify several dynamical aspects of the functional organization of the SN and DMN in the rat brain and help clarify conflicting findings in the literature.

AI stimulation of Chronos-expressing neurons in experimental rats, but not EYFP controls, induced robust ON and OFF brain states characterized by distinct patterns of activity and connectivity between the AI, PrL Cg, and multiple nodes spanning the anterior-posterior axis of the RSC. A comparison of these states revealed that AI stimulation suppressed neuronal responses in the RSC, and reductions were observed in multiple subdivisions of the RSC spanning its anterior-posterior axis (Fig. 4).

Importantly, our analyses also revealed heterogeneous activity changes in response to AI stimulation with a prominent role for the anterior to the middle, but not posterior, aspects of the RSC in the rat DMN (Fig. 4a, b). Convergent with our findings at the anterior RSC, Ferrier et al recently used single-slice functional ultrasound imaging in the mouse brain and reported that somatosensory sensory stimulation induces cerebral blood volume suppression in the anterior RSC[55]. Our results also parallel findings from Fakhraei et al. who measured local field potentials in multiple rat brain areas and found suppression of anterior RSC during a behavioral inhibition task[50]. Interestingly, suppression of the anterior RSC was linked with improved task performance, similar to findings previously reported in humans[56]. Our results showing higher activation levels of anterior RSC in the OFF state than the ON state is in agreement with these studies, and highlight the role of RSC activation and deactivation in the maintenance and suppression of the DMN. Crucially, these findings advance our knowledge by revealing a causal mechanism specifically tied to the direct feedforward stimulation of AI neurons.

AI stimulation also decreased functional connectivity between the AI and anterior RSC (Fig. 4c) as well as the decoupling of the anterior from posterior RSC (Fig. 4c), demonstrating a causal mechanism underlying the decoupling of the SN and DMN networks. Taken together, these results identify AI-induced focal suppression of RSC connectivity in rats which is consistent with observations of task-related decoupling of the SN and DMN in the human brain[2,4,7]. This finding represents a critical step towards the use of rodent models for understanding the circuit mechanisms of dynamic network switching processes observed in the human brain[2,8].

Direct stimulation of the AI provides clarification of circuit dynamics beyond resting-state fMRI, much like task-based activation studies and causal network analysis have provided in the context of human fMRI studies[8]. Pertinent here is the question of the functional organization and distinctness of SN and DMN in the rodent brain,

especially with respect to the PrL and Cg in the medial prefrontal cortex which has been variously assigned to both networks[17,32]. No clear distinctions from the viewpoint of the intrinsic functional network-level organization have emerged, and maps based on independent component analyses have emphasized SN and DMN overlap in both the PrL and Cg[23]. Indeed, while early studies of the DMN have suggested that the PrL and Cg might constitute its anterior node[17,20,32], anatomical tracer studies have pointed to direct pathways linking the PrL and Cg with the AI[17,57] a pattern supporting our findings here. Further evidence supporting the involvement of these regions in the SN was recently described by Tsai et al., showing increased AI-Cg functional connectivity during naloxone cue-induced conditioned heroin withdrawal[17].

In this context, it is noteworthy that AI stimulation selectively suppressed activity and connectivity with the RSC, but not the PrL or Cg. Instead, in parallel with decreases in AI connectivity with the RSC, the PrL showed increased connectivity with Cg and decreased connectivity with the posterior RSC (see also Supplementary Information: Discussion #8 for the heterogeneous relationship between activation profiles and functional coupling between the PrL and Cg). Thus, stimulation-related patterns suggest a state-specific alignment of the PrL and Cg more consistent with the SN, rather than the DMN. This finding should not be surprising as the PrL and Cg have been implicated in frontal control of memory, attention, and reward processing in the rodent brain[58]. The PrL has also been implicated in processing information from the environment, including drug cues[59,60]. Consistent with this view, a recent study using whole-brain fMRI in rats found that novelty preferences and cocaine-associated cues influence PrL association with the SN[61]. It is also possible that the functional distinctions of these medial frontal regions are not as well aligned into an SN-DMN dichotomy as observed in the human brain, consistent with reports of divergent rodent and primate medial frontal cortex functional connectivity[62]. Indeed, while chemogenetic suppression of Cg has been shown to reduce RSC activity and overall DMN connectivity[27,63], a recent study by Tu et al. showed that such manipulation also increases the centrality of RSC and promotes the emergence of the RSC hub of DMN[64]. Consistent with our findings, these results may suggest that Cg and PrL are involved in both SN and DMN, and these networks might be less segregated in rodents than in primates. Further research with manipulations of PrL and Cg with whole-brain fMRI is required to clarify circuit mechanisms of network organization and function across species.

An important question raised by our findings is whether AI stimulation induces DMN suppression via direct or indirect pathways. Our observation that AI stimulation modulates RSC activity and connectivity in the rat is intriguing given the lack of evidence for direct anatomical connections between the two structures in rodents[65-68]. Although it is tempting to assume that functional connectivity implies direct anatomical connectivity, it has long been established that the relationship between the two is far more complex; as such, even sparse indirect connections can be sufficient to induce changes in functional connectivity[69,70]. We suggest that information flow between AI and RSC may rely on one or more intermediary nodes in the mPFC. Indeed, rodent tract-tracing studies reveal that the AI is connected to Cg and PrL[17,19,67,71], which in turn are connected to the RSC via the cingulum bundle[67]. The PrL is also directly connected to Cg, and given that RSC has relatively dense connectivity with Cg compared to other prefrontal regions, a multi-node pathway for information flow from AI→PrL→Cg→RSC is also plausible. In addition, the claustrum may be an alternative intermediary node for AI-RSC connectivity, as it is bidirectionally connected with AI, prefrontal regions, and RSC[67], and recent evidence supports a role for this structure in SN-DMN interactions[72,73]. Characterizing the functional and causal role of these putative information flow pathways in SN–DMN dynamics is an important goal for future studies (see Supplementary Information: Discussion #9 for a

circuit-level explanation for RSC suppression by AI stimulation). Because the temporal resolution of fMRI limits inferences about the directionality of communication via intermediary nodes, electrophysiological studies may be required to delineate multi-node pathways.

Our computational analysis also revealed a short-lived, but stable, state that occurred during the transition between the OFF and ON states. Analysis of this state, which could not have been identified using conventional general linear models, revealed that transitions from the OFF to ON states likely occur by passing through this third, transition, state (Fig. 5). No such state was present in EYFP rats. Thus, our computational methods not only uncovered AI stimulation-induced brain states, but also transition states and their temporal boundaries. A particularly powerful feature of our modeling approach is therefore the ability to capture transition states which can carry information about short-lived changes that can be missed by conventional approaches. This allowed us to uncover several important facets of SN–DMN circuit dynamics.

Further analysis of the transition state revealed four features of activation and connectivity changes in the RSC associated with AI stimulation (Fig. 6). First, the transition state was characterized by strong suppression of anterior to middle RSC subdivisions in Chronos rats, with levels higher than those seen between OFF and ON states. Thus, this state captures a large change in fMRI response properties that occur during the transition from the OFF to ON states. Second, there was a transient increase in posterior RSC activity with AI stimulation. Third, during the transition state, AI connectivity with a specific middle RSC subdivision increased above levels observed in the OFF state. Fourth, the transition state was characterized by decreased connectivity between anterior and middle RSC subdivisions. Collectively, these results demonstrate that dynamic state changes induced by AI stimulation in the rat brain begin with a heterogenous pattern of node-specific transient suppression along the AP axis of the RSC, in parallel with short-lived subregion-specific AI-RSC coupling and intra-RSC decoupling. These findings further reveal heterogeneous profiles of changes in functional circuits and hint at inhibitory processes that may serve to transiently create distinct functional boundaries within the RSC such as those observed during various stages of memory formation in the rodent brain[16]. Taken together, these findings suggest that previous studies, which have almost always focused on the predefined task/block boundaries to examine functional connectivity, are likely to miss key features of brain state dynamics and the unique functional circuits associated with them.

The RSC is involved in a variety of cognitive tasks including memory, navigation, and prospective thinking, yet the precise role of the rodent RSC and the functional differences between its subdivisions remain elusive[25]. Our analyses revealed AI-stimulation-induced heterogeneity within RSC consistent with recent reports about its functional organization along the anterior-posterior axis[16]. While rodent resting-state fMRI studies have treated the RSC as a homogeneous region, electrophysiological and optogenetic studies suggest that the RSC contains multiple cytoarchitectonic divisions[16,25]. Consistent with our findings, optogenetic inhibition of neuronal activity in the anterior and posterior RSC revealed divergent impacts on behavior evoked by conditioned stimuli[16]. Taken together, these findings suggest that the RSC comprises distinct functional modules, consistent with our finding of clear differences in connectivity patterns associated with the stimulation of AI. Further studies with behaviorally relevant salient stimuli in awake rodents, whole-brain imaging, and causal circuit manipulations, including inhibition of the AI, mPFC nodes, and RSC, will advance our understanding of RSC functional heterogeneity (see Supplementary Information: Discussion #10 for the importance of replication experiments in awake animals). It is noteworthy here that heterogeneity along the anterior/posterior axis of the posterior cingulate cortex is also a prominent feature of the human DMN[74], and

optogenetic manipulations with whole-brain imaging have the potential to inform the underlying causal circuit mechanisms.

In summary, our study demonstrates that feedforward optogenetic stimulation of the AI suppresses activation of the DMN and dynamically decouples SN and DMN. These findings parallel reports of decreased connectivity between SN and DMN observed during working memory and other cognitively demanding tasks in humans[2,4,7]. Notably, direct stimulation of neurons allowed us to address limitations of prior noninvasive studies and probe the direct effects of manipulation of neuronal activity in the AI. Our study elucidates causal mechanisms underlying neuronal perturbation of cross-network interactions involving the SN and DMN and converges on previous reports in human fMRI studies implicating the AI in network switching[8]. Our findings also suggest that PrL and Cg, commonly included in the rodent DMN, may also have a functional role in the SN, pointing to lower levels of network segregation in rodents than in primates. More generally, our dynamical systems approach and findings address critical gaps in our understanding of the cellular organization of brain networks[75,76] and provide a translational model for investigating aberrant network switching in psychiatric disorders[11,77]. Finally, our innovative computational tools and analytic approach will be useful for investigations of latent processes underlying brain function and dysfunction.

## Methods
### Data acquisition
**Surgical preparation of rats for optogenetic manipulation.** Sixteen, male, Sprague Dawley rats (Charles River Laboratories, Wilmington, MA, USA), ~60 days old and weighing ~300 g at the time of surgical preparation, were separated into experimental (Chronos, $n = 9$) and control (EYFP, $n = 7$) groups for these studies. Given the reported sex-specific differences in rodent responses to anesthesia[78–81] and the lack of characterization of such differences in the anesthesia protocol used in this study for fMRI scanning (as described below), only male rats were utilized in this investigation.

During stereotactic surgery, rats were isoflurane-anesthetized (4% induction, 2% maintenance) and all procedures were conducted using sterile techniques. After exposing the skull, a craniotomy was drilled above the right AI and a 30 G infusion cannula preloaded with the virus was slowly lowered into the right AI at the following coordinate from the skull at Bregma according to the Paxinos and Watson rat brain atlas, 6th edition: 2.75 mm AP, 3.75 mm medial-lateral, −6.00 mm dorsal-ventral. After allowing the brain tissue to settle, the cannula was raised 0.1 mm and 1 μl of AAV5-hSyn-Chronos at a flow rate of 0.1 was infused at a flow rate of 0.1 μl/min into the right AI of Chronos rats. Following the virus infusion, ten minutes were given to allow virus diffusion before the infusion cannula was retracted.

Chronos was chosen for selective feedforward control of AI neurons because of its superior spiking fidelity[82] and efficiency in recruiting axonal feedforward activity during repetitive stimulation compared to conventional channelrhodopsin-2[83]. The pan-neuronal hSyn promoter was chosen for its high transduction efficiency in cortex[84], and because further neuronal specificity was not required as the AI sends exclusively excitatory projections to other brain regions[71]. Control rats received the same preparation, but with an AAV5-hSyn-EYFP control vector to express only EYFP instead of Chronos. The required vectors were prepared and packaged with a titer of ~$10^{12}$ vg/ml by the UNC Vector Core at UNC Chapel Hill.

During the same surgery, following virus infusion, an optical fiber (200 μm core, 0.22 NA) was implanted just above the infusion site at DV −5.70 mm to enable light delivery to the right AI for optogenetic stimulation. Thereafter, 3 or 4 MR-compatible brass screws were secured into the skull near the lateral ridges without damaging brain tissue. Dental cement was then spread over the exposed skull surface

to firmly secure the optical fiber in place with MR-compatible brass screws serving as anchor points. Surgical implantation accuracy of optical fibers was verified with anatomical MRI scans (T2-weighted RARE sequences) at the beginning of fMRI sessions (Fig. 1c)[14,85].

Following fMRI scanning, subjects were deeply anesthetized and transcardially perfused with saline, followed by 4% paraformaldehyde, to affix brain tissue for histological verification of viral vector expression. Brains were postfixed in 4% paraformaldehyde for 24 h and transferred to 30% sucrose for 48 h before sectioning into 40 μm coronal slices by a cryostat. Sections spanning the AI were then mounted and coverslipped with Fluoroshield with DAPI (Sigma Aldrich). Finally, mounted brain sections were visualized using an Olympus MVX10 wide field scope with a 5× objective to confirm virus expression in AI, and obtain representative images.

All animal procedures were performed in strict accordance with the National Institutes of Health Guidelines for Animal Research (Guide for the Care and Use of Laboratory Animals, eighth edition) and reviewed and approved by the University of North Carolina Institutional Animal Care and Use Committee (protocol #15-057.0).

**Preparation of rats for fMRI scanning.** Approximately, 30 min prior to fMRI scanning, rats were anesthetized with 2% isoflurane in medical air (4% for induction), orotracheally intubated and mechanically ventilated, and outfitted with an intravenous tail-vein catheter. Next, rats were secured in a custom MRI cradle, and a fiber optic patch cable (200 μm core, 0.22 NA, 7 m long), set to deliver blue light from a solid-state laser, was connected to the implanted fiber in each rat for optogenetic stimulation of the AI. Subsequently, rats were positioned in the MRI bore, anatomical scans were completed, then a functional scan was acquired during which the CBV contrast agent Feraheme (30 mg/kg intravenous) was administered to augment detection sensitivity and measure CBV-weighted signal changes[41,86–88]. Finally, rats were switched to a well-established sedation protocol combining low-dose (0.5%) isoflurane with a cocktail of intravenous dexmedetomidine (0.05 mg/kg/h) and pancuronium (0.5 mg/kg/h) and allowed 30 min for their physiological parameters to stabilize before optogenetic-fMRI scanning. Importantly, this light sedation protocol has been shown to preserve neurovascular coupling for functional connectivity studies[89–94]. Throughout fMRI scanning procedures, animal physiology was continuously monitored and maintained within physiologically desirable limits, including a $3.0 \pm 0.3\%$ end-tidal $CO_2$, 250–320 bpm heart rate, $93 \pm 5\%$ oxygen saturation ($SpO_2$), and $37.5 \pm 1 °C$ core temperature.

**MRI acquisition protocols.** fMRI studies were performed on a Bruker 9.4-Tesla/30-cm scanner with a BGA9-S gradient insert, operated with the ParaVision 5 preclinical imaging software suite, at the Center for Animal Magnetic Resonance Imaging at UNC Chapel Hill. A homemade surface coil (1.6 cm inner diameter), designed to not obstruct optical fibers, was used as the RF transceiver. Magnetic field homogeneity was optimized by global shim followed by local first- and second-order shims using the standard FASTMAP protocol[95]. All MRI acquisitions were in anisotropic resolution with 12 coronal slices, 1 mm thick and 1 mm apart, aligned on the AP axis such that the fifth-most anterior slice was centered on the anterior commissure (corresponding to −0.36 mm AP) for each rat. fMRI scans were acquired with a single shot, gradient echo-EPI sequence optimized for the Feraheme CBV contrast agent with the following parameters: spectral width = 300 kHz, TR/TE = 1000/8 ms, FOV = $2.56 \times 2.56$ cm$^2$, matrix size = $80 \times 80$ (Fig. S1). Anatomical images were acquired using a T2-weighted RARE sequence with the following parameters: spectral width: 47 kHz, TR/TE = 2500/33 ms, FOV = $2.56 \times 2.56$ cm$^2$, matrix size = $256 \times 256$, RARE factor = 8, averages = 8. The initial Feraheme infusion fMRI scan was 300 s in duration and included a minimum of 60 s before and after the infusion to acquire the pre- and post-Feraheme image intensities,

respectively. CBV contrast was confirmed for each rat by comparing pre- and post-Feraheme image intensities. Optogenetic-fMRI scans were acquired in a single 1160 s run for each animal, beginning with a stimulation OFF resting period for the initial 360 s followed by an 800 s period of repeated stimulation blocks described below.

**Optogenetic stimulation.** Stimulation was delivered in a repeated epoch design, consisting of 8 repetitions of a 20 s "ON" block of pulsed 473 nm blue-light delivery via the chronically-implanted optical fiber in AI followed by an 80 s "OFF" block without stimulation to allow recovery of neuronal activity to baseline. Based on the extant literature[41,96], stimulation light pulses were delivered at 20 Hz (10 mW power at fiber tip, 5 ms pulse width).

### Data analysis
**fMRI preprocessing.** All MRI images were preprocessed using AFNI (http://afni.nimh.nih.gov/afni/, ver.20.2.10). Briefly, a group template based on individual T2-weighted images was generated using three subsequent steps: (1) transformation and linear deformation, (2) nonlinear deformation, and (3) averaging[97]. Individual subject fMRI data were then slice-timing corrected and realigned to the mean image to correct head motion. Six degrees of freedom motion parameters were estimated through the realignment step. Aligned images were then co-registered to the T2-weighted anatomical scan followed by spatial normalization to the T2-weighted group template using linear affine registration. The normalized functional images were resampled from $0.32 \times 0.32 \times 1$ mm voxel size to 0.5 mm isotropic voxel size. Nuisance removal includes detrending with 3rd-order polynomial fitting, a high-pass filter with a frequency cut at 0.01 Hz, and regressing out the head motion parameters. Finally, a 1 mm FWHM Gaussian kernel was applied for spatial smoothing.

**ROI time series.** To test our hypotheses related to dynamic changes in the SN and DMN induced by optogenetic stimulation of the AI, and to avoid circularity arising from bias in selection[98], we used nine anatomically-defined canonical SN and DMN nodes encompassing the AI, PrL, Cg, and RSC[16,17,20–28]. The precise locations of these nodes were based on a wide range of published studies in rodents[16,17,20–28]. Because of its relatively large size, and growing evidence for functional heterogeneity of the RSC along its anterior/posterior axis[16], we demarcated six ROIs along the A/P axis of the RSC. Spherical ROIs 1 mm in diameter were placed at the center of mass within the anatomical boundaries of each region based on the Paxinos and Watson rat brain atlas, 6th edition[49], after co-registration to group-template image space. Table 1 shows the coordinates of each ROI used in the main analyses. Next, CBV-weighted time-series data were extracted from each ROI (noncontiguous bilateral ROIs were averaged) by inverting raw fMRI signal following intravenous administration of Feraheme and displayed in percent raw signal changes[86]. Finally, time courses were normalized for use in computational modeling and group comparisons. Control analyses were conducted using anatomically-defined ROIs in the (i) amygdala, hippocampus, (ii) striatum and MTL, and (iii) auditory, visual, and motor cortices (Figs. S11a–S13a). For display purposes, CBV responses to optogenetic stimulation were expressed as a percent change from pre-stimulation baseline values. Amplitudes were calculated as mean percent CBV changes during stimulation epochs.

**Bayesian switching dynamic systems model.** We used BSDS, based on a hidden Markov process model, to uncover latent brain states associated with the optogenetic stimulation of AI.

We used a variational Bayesian (VB) framework to infer model parameters, including the number of brain states. The number of states is treated as a random variable, whose optimal value is learned from data using automatic relevance determination procedures implemented in a VB framework[43]. BSDS models were initialized with 10 states for both Chronos and EYFP rats. BSDS identified 5 latent brain states in the Chronos rats and 3 latent brain states in the EYFP rats. A detailed explanation of the model is in Supplementary Information: Materials and Methods. Key measures extracted from BSDS include posterior probability, occupancy rate, and covariance of latent brain state.

### Reporting summary
Further information on research design is available in the Nature Portfolio Reporting Summary linked to this article.

## Data availability
All original data reported in this study are publicly available on Zenodo: https://zenodo.org/badge/latestdoi/434680845[99]. Source data are provided in this paper.

## Code availability
The code used to perform the analyses in this study are publicly available on Zenodo: https://zenodo.org/badge/latestdoi/434680845[99].

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

## Acknowledgements

This work was supported in part by the National Institute of Mental Health (R01MH121069 to V.M., and R01MH126518, RF1MH117053, R01MH111429, S10MH124745 to Y.-Y.I.S.), National Institute on Alcohol Abuse and Alcoholism (P60AA011605 and U01AA020023 to Y.-Y.I.S., T32AA007573 to D.C.), National Institute of Neurological Disorders and Stroke (R01NS086085 to V.M., R01NS091236 to Y.-Y.I.S.), National Institute of Child Health and Human Development (P50HD103573 to

Y.-Y.I.S.), National Institute of Biomedical Imaging and Bioengineering (RO1EB022907 to V.M.), and National Institute of Health Office of the Director (S10OD026796 to Y.-Y.I.S.). We thank members of the UNC Center for Animal MRI (CAMRI) for technical assistance and Drs. Harry Chao and Li-Ming Hsu for insightful discussions.

## Author contributions

V.M., D.C., and B.L. contributed equally to this work. V.M., D.C., B.L., S.-H.L., and Y.-Y.I.S. conceptualized the study. D.C. performed animal surgeries, fMRI scanning, and histology. D.C., R.Y., B.L., and S.-H.L. preprocessed the raw fMRI data. B.L. implemented the BSDS model. V.M., D.C., B.L., R.Y., and S.-H.L. analyzed the data. V.M., D.C., B.L., and Y.-Y.I.S. interpreted the data and wrote and revised the paper. All authors read and approved the final paper.

## Competing interests

The authors declare no competing interests.
