## [Peer Review File · Nature Communications]

Optogenetic stimulation of anterior insular cortex neurons reveals causal mechanisms underlying suppression of the default mode network by the salience networkREVIEWER COMMENTS

Reviewer #1 (Remarks to the Author):

This excellent study by Menon et al. demonstrates the power of optogenetic stimulation in combination with fMRI for interrogating the functional organization of the brain. The work is thorough and rigorous, incorporating both experimental and analytical controls. The manuscript could be published as it is, but I have a few requests that might enhance its value for future readers.

1) Please show some of your raw EPI data in the supplemental material if at all possible. Image quality can vary greatly across studies, and showing a typical image volume helps the reader to assess where signal dropout or distortion can affect the results.

2) Part of the power of fMRI is its capability to map the entire brain, which sometimes identifies unexpected areas of interest. While I appreciate the hypothesis-driven selection of ROIs for this study, it would be interesting to see the whole-brain maps of activity for each brain state. A simple average of the time points assigned to each state should suffice.

3) Another benefit of including whole brain images for each brain state is that the reader can visually compare the brain states not just within a group, but across groups. I would really like to know how the spatial patterns of activity in the brain states for the Chronos group compare to the control group. One would expect the primary OFF state in the Chronos group to be the primary state in the control group. This would also allow the investigation of why there are 3 states in the control group and 5 in the Chronos group. If three of the states look the same for both groups, that would suggest the other two are driven by the stimulation.

4) I am also curious about the spatial distribution of signal in the transition state. It looks rather flat across ROIs, which suggests it might have a relatively even distribution across the whole brain. These flat states are common in HMMs (see for example Fig. 3 in <https://pubmed.ncbi.nlm.nih.gov/27008543/>) and I wonder if they are a reflection of the fact that mirrored brain states are often obtained, and the signal must pass through zero when switching from positive to negative. I'm not completely convinced of the importance of this state specifically for ON and OFF transitions, since it also appears to mediate transitions between other states (e.g., 5 and 4 in Fig. 5).

5) Minor point 1: In Fig. 1d, the percent change is very small, and the baseline appears to go below zero. Is this related to features of the hemodynamic response to optogenetic stimulation?

6) Minor point 2: These data were obtained in anesthetized animals. Given that the networks examined are involved in complex behaviors including attention, reward-seeking, etc., the dynamics may be very different in awake animals, which merits a few sentences in the discussion.

Shella Keilholz

Reviewer #2 (Remarks to the Author):

This study examined the effect of feedforward optogenetic stimulation of the anterior insular cortex (AI) on dynamic functional connectivity with key nodes of the salience network (SN) and default-mode network (DMN). A main finding was that stimulation of the AI was sufficient to suppress key nodes of the DMN. This is an important study that provide novel insight into causal mechanisms underlying brain network switching.

The authors selectively applied optogenetic stimulation of chronos-expressing neurons within the right AI. Could the authors further elaborate on what percentage of neurons within the AI express chronos, are these chronos-expressing neurons found within specific cortical layers (input vs output layers) of the AI and is it selectively expressed within excitatory neurons? Further, the AI is often subdivided into agranular, dysgranular and granular subregions. Could the authors be more specific whether the viral injection/optogenetic manipulation targeted a specific AI subregion? Why did the authors choose to selectively target the right AI rather than randomly targeting both the left and right AI? Would it be

expected that similar findings were obtained if the left AI had been targeted? Finally, I assume that the SN and DMN ROIs were also all on the right side.

The Discussion section addressed to some extent whether the present findings are supported by anatomical evidence. The authors mentioned that all AI output neurons are excitatory neurons, thus the finding that AI stimulation suppresses DMN network activity might suggest that these AI output neurons (or intermediate regions) target inhibitory neurons. Would it be possible to elaborate on this topic in the discussion?

Do the authors think that the present findings that AI activation causally suppresses DMN activity is sufficient to fully explain the functional antagonistic role between the SN and DMN networks, or do the authors think that DMN activation might also directly suppress SN activity? Further, would the authors think that optogenetic inhibition of the AI might be sufficient to induce an activation of the DMN?

The authors further discuss a role of the prelimbic cortex (PrL) in both the SN and DMN. Is it possible that the conflicting evidence here is also caused by an anatomical/functional heterogeneity? Work by Arnsten and colleagues has shown that particularly the more anterior part of the PrL is involved in working memory and receives most noradrenergic innervation.

Reviewer #3 (Remarks to the Author):

The manuscript is of substantial interest, as it concerns causal manipulations triggering brain state changes related to the DMN. As such, it is not the first to use this approach, and previous work in this area (PNAS 115 (6) 1352-1357, SCI REP; 9: 2570. and others) should be mentioned and discussed in relation to the present findings. This above mentioned work concerns long range corticopetal projections that activate or deactivate the DMN, rather than interfering directly in the cortex. The previous work mostly focused on behavioral and electrophysiological changes, so the present study is to my knowledge one of the first to combine a causal approach with whole brain fMRI, which is highly commendable, but other work exists here too, eg iScience 23(9):101455. It would be useful to add a section in the discussion as to what degree the present findings are consistent with these previous studies.

On a general level, the work appears to be well executed, although at times I feel it could be a little more coherently organized (see also below). The scientific bottom line of subspecialisation of PRL in the rat is interesting, as is the relation to the cingulate. Reading the manuscript, I was wondering if some of these findings might be related to neuroanatomical data on area connection strength for example (if such work exists in the literature regarding the AI cortex (where are its major outputs going to). It would be of interest to see a short section in the discussion in this regard.

The points below occurred to me as I was reading the manuscript, I realize that some of them are addressed actually later, but maybe the authors find this information useful for streamlining the manuscript.

L.178: The authors document expression of Chronos-YFP and YFP in the anterior insula, but I did not see any details about which cell types these were expressed in. Histological analyses would be useful here, or pointing to data from publications that have employed the same viral vector and described targeted cell types.

L.199: The description here along with the fig.2 does not appear to detail any findings, but explains the procedure. This might be stated explicitly, or this information moved to supplement.

L.208: According to what criteria were 5 brain states defined? I expect that there is also probably some divergence among the results of the individual rats. Please elaborate. It should be mentioned that other states will be discussed later.

L.224: There do not seem to be any error bars on the 50% SVM classification estimate.

L.252: The other 3 states are not mentioned further, but authors focus only on states 1 and 2. Coming back to my point above, why were 5 states chosen as cutoff in the first place, and are results identical if 2 states are chosen. In general, have the authors analyzed the data using more standard fMRI methods, such as for example GLM? What are the results of such an analysis; these should be able to capture the same effects as reported here.

L.395: The authors observe PRL activation in some segments in ON blocks, as well as increased PRL-CG coupling on ON blocks but no activation of CG on ON blocks. How is this possible? This is an interesting finding; it is worth discussing in a little detail how these 3 effects can occur at the same time. This discussion is actually presented in L.450, it would seem that streamlining the discussion might be helpful to the reader.

L.495: The material related to the transition states seems speculative and does not provide any major advances in understanding; focusing the manuscript on the major discoveries might be advantageous.

Re: NCOMMS-22-25913A-Z, Optogenetic stimulation of anterior insular cortex neurons reveals causal mechanisms underlying suppression of the default mode network by the salience network

Reviewer #1

[Overall Comments] This excellent study by Menon et al. demonstrates the power of optogenetic stimulation in combination with fMRI for interrogating the functional organization of the brain. The work is thorough and rigorous, incorporating both experimental and analytical controls. The manuscript could be published as it is, but I have a few requests that might enhance its value for future readers.

[Response] We thank the reviewer for the positive comments regarding the significance of our work and recommending our manuscript for publication in *Nature Communications*. We appreciate the constructive comments and suggestions, which we have incorporated into the revised manuscript.

[Comment #1.1] Please show some of your raw EPI data in the supplemental material if at all possible. Image quality can vary greatly across studies, and showing a typical image volume helps the reader to assess where signal dropout or distortion can affect the results.

[Response] We now show a representative image volume from our raw EPI data in the *Supplemental Information: Figures* section (**Figure S1**, line 725; **Figure R1** below), and direct readers to this figure on line 163 and 630 of the main text.

Figure R1. A representative image volume from raw, CBV-contrast-enhanced EPI data. Data correspond to the first volume of the fMRI scan with AI-stimulation for a Chronos rat. Coronal slices are displayed posterior to anterior, from top-left to bottom-right. D = dorsal, V = ventral, L = left, R = right.

[Comment #1.2] Part of the power of fMRI is its capability to map the entire brain, which sometimes identifies unexpected areas of interest. While I appreciate the hypothesis-driven selection of ROIs for this study, it would be interesting to see the whole-brain maps of activity for each brain state. A simple average of the time points assigned to each state should suffice.

[Response]. We thank the reviewer for this insightful suggestion. Following the reviewer’s suggestion, we averaged time points assigned to each brain state and visualized the whole-brain maps of activity for each brain state in the Chronos group (**Figure R2**). Briefly, this analysis demonstrated different patterns of activation and deactivation in the AI and RSC across brain states. Notably, we observed activation of the RSC spanning anterior-to-middle subdivisions from -3.08 to -4.28 and deactivation of the AI in State 1, the OFF-state, and activation of the AI in State 2, the ON-state, which corroborate our results in **Figure 4A**. In addition, we observed deactivation of anterior-middle RSC and AI in State 3, the Transition state, which corroborates our results in **Figure 6A, D**, respectively. We have included this new result and figure in *Supplemental Information: Results and Figures* sections (line 205~225, **Figure S4A** at line 791), and direct readers to this information on line 202~204 of the main text.

Figure R2. State-specific brain activation and deactivation patterns (z-maps) in the Chronos group. Whole brain maps for each brain state were determined by averaging time points assigned to the respective state.

[Comment #1.3] Another benefit of including whole brain images for each brain state is that the reader can visually compare the brain states not just within a group, but across groups. I would really like to know how the spatial patterns of activity in the brain states for the Chronos group compare to the control group. One would expect the primary OFF state in the Chronos group to be the primary state in the control group. This would also allow the

investigation of why there are 3 states in the control group and 5 in the Chronos group. If three of the states look the same for both groups, that would suggest the other two are driven by the stimulation.

[Response] We thank the reviewer for this excellent suggestion. As suggested, we averaged time points assigned to each brain state and plotted the whole-brain maps of activity for each brain state in the EYFP control group (**Figure R3**). We then computed the spatial correlation coefficients of respective states from the Chronos and EYFP groups to examine the similarity between them. The spatial patterns of State 1, the OFF-state, in the Chronos group (**Figure R2**), and State 1, the primary state, in the EYFP control group (**Figure R3**) showed high similarity ($r=0.47$), suggesting that the primary OFF state in the Chronos group may be the primary state in the control group as predicted by the reviewer (**Figure R4**). Furthermore, State 2, the ON-state, and State 3, the transition-state, in the Chronos group showed high similarity with State 3 and State 2 in the EYFP control group, respectively. This result suggests that the other two states in the Chronos group may be driven by internal events. We have now included these new results and figures in *Supplemental Information: Results and Figures* sections (line 205~225, **Figure S4** at line 791), and direct readers to this information on line 202~204 of the main text.

Figure R3. State-specific brain activation and deactivation patterns (z-maps) in the EYFP control group. Whole brain maps for each brain state were determined by averaging time points assigned to the respective state.

Figure R4. Spatial correlation between brain states identified in the Chronos and EYFP control groups.

[Comment #1.4] I am also curious about the spatial distribution of signal in the transition state. It looks rather flat across ROIs, which suggests it might have a relatively even distribution across the whole brain. These flat states are common in HMMs (see for example Fig. 3 in <https://pubmed.ncbi.nlm.nih.gov/27008543/>) and I wonder if they are a reflection of the fact that mirrored brain states are often obtained, and the signal must pass through zero when switching from positive to negative. I'm not completely convinced of the importance of this state specifically for ON and OFF transitions, since it also appears to mediate transitions between other states (e.g., 5 and 4 in Fig. 5).

[Response] We thank the reviewer for the valuable feedback and suggestions. The reviewer is correct in that compared to the ON and OFF state, the transition state looks rather flat across the whole brain (**Figure R2**) similar to the ones noted by Chen *et al*¹. Furthermore, these flat states do appear to be a reflection of mirrored brain states, such that the signal must pass through zero when switching from positive to negative. However, it should be noted that in our model brain states are not defined by activation alone, rather both activation and connectivity differences underlie state differences.

The reason we investigated a role of State 3 specifically in relation to State 1 (i.e., OFF-state) and State 2 (i.e., ON-state) is because these two states showed dominant occupancy rates during the OFF and ON stimulation blocks, respectively (**Figure R5A**). As detailed in the *Supplemental Materials: Results* section (line 299-323), this result naturally led us to explore a dynamical transition process between OFF and ON states at the stimulation boundaries (**Figure R5B, left**). Consequently, our analysis demonstrated that direct transition between OFF and ON states are rare ($P_{OFF \rightarrow ON} = 0.01$, $P_{ON \rightarrow OFF} = 0.01$). Rather, the transitions occurred by first passing through State 3 ($P_{OFF \rightarrow State3} = 0.28$, $P_{State3 \rightarrow ON} = 0.20$, $P_{ON \rightarrow State3} = 0.11$, $P_{State3 \rightarrow OFF} = 0.18$, **Figure R5B**), suggesting a role for State 3 as a Transition state between OFF and ON states during the stimulation boundaries. While State 3 also occasionally mediated transitions between States 4 and 5, these two states not only have the lowest occurrence rates during both ON and OFF stimulation blocks (**Figure R5B**), but also showed volatility from one time-step to another (i.e., the probability of persisting at its own state is lower than the probabilities of switching to other states). These results suggest that States 4 and 5 are highly volatile and noisy.

Figure R5. Temporal properties of latent brain states in Chronos rats. (A) Figure 3D from the manuscript. Occupancy rates of latent brain states in Chronos rats. * $p < 0.05$, two-tailed t-test, FDR-corrected. **(B)** Figure 5A from the manuscript. State switching probability of latent brain states at stimulation boundaries identified in Chronos rats.

[Comment #1.5] Minor point 1: In Fig. 1d, the percent change is very small, and the baseline appears to go below zero. Is this related to features of the hemodynamic response to optogenetic stimulation?

[Response] We thank the reviewer for this feedback. The baseline was below zero because of slow signal drift over time. We have now normalized the baseline to be zero. CBV responses are expressed as a percent change from pre-stimulation baseline values. Amplitudes were calculated as mean percent CBV changes during stimulation epochs.

[Comment #1.6] Minor point 2: These data were obtained in anesthetized animals. Given that the networks examined are involved in complex behaviors including attention, reward-seeking, etc., the dynamics may be very different in awake animals, which merits a few sentences in the discussion.

[Response] We agree and have included the following statement about the potential effects of our anesthesia protocol and the desirability of replication experiments in awake animals in the *Supplementary Information: Discussion* section (line 656~665), with reference to this additional discussion in the main text on line 534~535.

“We employed a well-established, low-dose, dexmedetomidine/isoflurane anesthesia protocol²⁻⁵ to optogenetically interrogate AI activation effects with minimal contaminations from background neuronal activity or DMN suppression by salient stimuli. Although static FC networks show good correspondence between this protocol and awake conditions^{6,7}, and stimulus-evoked responses are robust⁴, more profound differences have been observed in dynamic FC⁸, especially for networks related to complex behaviors, attention, and cognitive function like the DMN and SN⁹. Further work with awake animals is needed to validate and replicate our findings.”

Reviewer #2

[Overall Comments] This study examined the effect of feedforward optogenetic stimulation of the anterior insular cortex (AI) on dynamic functional connectivity with key nodes of the salience network (SN) and default-mode network (DMN). A main finding was that stimulation of the AI was sufficient to suppress key nodes of the DMN. This is an important study that provide novel insight into causal mechanisms underlying brain network switching.

[Response] We are pleased that the reviewer found our study and findings to be important. We found the reviewer’s comments to be insightful; addressing them in detail, as noted below, has helped us further improve our manuscript.

[Comment #2.1] The authors selectively applied optogenetic stimulation of chronos-expressing neurons within the right AI. Could the authors further elaborate on what percentage of neurons within the AI express chronos, are these

chronos-expressing neurons found within specific cortical layers (input vs output layers) of the AI and is it selectively expressed within excitatory neurons?

[Response] We achieved Chronos and Control vector gene expression in the right AI via high-titer ($\sim 10^{12}$ vg/ml) recombinant adeno-associated virus (AAV) pseudo-typed with the serotype 5 capsid and using the human synapsin (hSyn) promoter. The pan-neuronal hSyn promoter was chosen for its high transduction efficiency in cortex¹⁰, and because further neuronal specificity was not required as the AI sends exclusively excitatory projections to other brain regions¹¹. We observed virus expression throughout all layers of the AI (**Figure 1B**), indicating expression in both inhibitory interneuron populations and excitatory output neurons. This broad expression across cortical layers and neuron types has been reported previously for this serotype¹² and promoter^{10,12,13}. Importantly, it has been demonstrated that optogenetic stimulation produces significantly increased multi-unit activity in rodent mPFC using the same serotype, promoter, stimulation frequency, pulse width, and a similar laser power and viral titer as employed here¹⁴.

This elaboration on the expected Chronos expression profile can now be found in the *Supplementary Information: Results* section (line 165~185), and is referred to in the main text on lines 165~167 and 169~171.

[Comment #2.2] Further, the AI is often subdivided into agranular, dysgranular and granular subregions. Could the authors be more specific whether the viral injection/optogenetic manipulation targeted a specific AI subregion?

[Response] We targeted the approximate center of mass of the AI, considered to be anterior to the granular and posterior insular cortex subdivisions¹⁵, for optogenetic stimulation. As such, fiber locations were within the dorsal and ventral agranular insular cortex (**Figure R6**), although some light spread could have activated Chronos-expressing neurons in the dysgranular insular cortex as well (**Figure R6**). Notably, although our fiber locations were anterior to granular and posterior insular cortex subdivisions, given the extensive reciprocal connectivity between insular cortex subdivisions¹¹, we cannot rule out network-level effects from these areas.

We have updated **Figure 1, Panels B and C**, and the figure caption (line 975~976, line 978) to include these subdivisions, as shown in **Figure R6**. The above discussion on targeted subregions can now be found in the *Supplementary Information: Results* section (line 165~185), and is referred to in the main text on lines 165~167 and 169~171.

Figure R6. Updated histology figure panels. Virus spread was present in the dorsal agranular insular cortex (AID), ventral agranular insular cortex (AIV), and dysgranular insular cortex (DI). Optical fibers were located within the AID and AIV.

[Comment #2.3] Why did the authors choose to selectively target the right AI rather than randomly targeting both the left and right AI? Would it be expected that similar findings were obtained if the left AI had been targeted? Finally, I assume that the SN and DMN ROIs were also all on the right side.

[Response] Our choice of targeting the right AI is based on previous findings about lateralization of AI dynamic causal influence on network switching¹⁶. Specifically, recent dynamic causal analysis of human fMRI data indicated that the right AI has significantly higher net causal outflow than left AI and is likely to be responsible for dynamic network switching between SN and DMN in healthy subjects¹⁷. In agreement, damage to the right AI in disease states impairs this dynamic network interaction¹⁸⁻²¹. Based on these findings, and potential for translational neuroscience research, we targeted the right AI and investigated the causal influence of AI stimulation on putative DMN nodes. We have now added this additional rationale in the *Supplementary Information: Discussion* section (line 462~472), referred to on line 163~164 of the main text. We predict similar effects with left AI stimulation in rodents, since the rodent brain shows less hemispheric specialization²². Finally, as shown in **Table 1**, only the AI ROI was right lateralized and we extracted time-series of PrL, Cg, and RSC subdivisions from bilateral ROIs given their midline locations (**Figure R7**).

Figure R7. Region of interests (ROIs). SN and DMN regions of interests (ROIs) used in this study. Note that only the AI stimulation site was right lateralized.

[Comment #2.4] The Discussion section addressed to some extent whether the present findings are supported by anatomical evidence. The authors mentioned that all AI output neurons are excitatory neurons, thus the finding that AI stimulation suppresses DMN network activity might suggest that these AI output neurons (or intermediate regions) target inhibitory neurons. Would it be possible to elaborate on this topic in the discussion?

[Response] We thank the reviewer for this suggestion. We have now added the following discussion of potential explanations for our findings given the anatomical evidence available to date in the *Supplementary Information: Discussion* section (line 634~653). This new supplementary information is referred to in the main manuscript on lines 486~487.

“At least four parsimonious, non-mutually exclusive, circuit-level explanations could account for RSC inhibition by stimulation of excitatory AI output. (1) AI could send direct projections to inhibitory interneurons in RSC which can inhibit local principal neurons via feedforward inhibition²³, but robust AI-RSC projections have not been shown in the tract tracing literature. (2) AI output could enhance activity of intermediary nodes (e.g. PrL or Cg) that in turn send excitatory projections to RSC²⁴, and these projections could preferentially drive RSC inhibitory interneurons, or (3) AI output could preferentially target inhibitory interneurons in intermediate nodes, thereby reducing excitatory drive from these nodes to RSC principal neurons. However, the cell-type specific targets of AI to intermediary node projections or intermediary node to RSC projections have not yet been characterized. (4) RSC subdivisions share dense reciprocal connections²⁵⁻²⁷, and if these connections preferentially target interneurons, then activation of one subdivision as a direct or indirect result of AI stimulation could inhibit other subdivisions. While this explanation could also contribute to RSC heterogeneity²⁸, described in the literature and corroborated by our findings, RSC heterogeneity could be accounted for by differential input from other brain areas. Further anatomical and neural dissection of these putative pathways should reveal critical anatomical circuit-level interactions underlying SN-DMN network interactions.”

[Comment #2.5] Do the authors think that the present findings that AI activation causally suppresses DMN activity is sufficient to fully explain the functional antagonistic role between the SN and DMN networks, or do the authors think that DMN activation might also directly suppress SN activity? Further, would the authors think that optogenetic inhibition of the AI might be sufficient to induce an activation of the DMN?

[Response] We appreciate the reviewer’s questions. Our findings demonstrate that AI activation causally suppresses DMN activity and is one likely mechanism underlying the functional antagonistic role between the SN and DMN networks. This, however, does not rule out other mechanisms including the possibility that the DMN activation might also directly, perhaps subsequently, suppress SN activity. Although no direct evidence exists yet, we do think that optogenetic inhibition of the AI might be sufficient to induce an activation of the DMN. Characterizing the differential causal role of SN and DMN nodes in this regard is an important goal for future studies. We have now included this discussion point in the revised *Supplementary Information: Discussion* section (line 555~563), which is referred to on lines 388~389 of the main manuscript.

[Comment #2.6] The authors further discusses a role of the prelimbic cortex (PrL) in both the SN and DMN. Is it possible that the conflicting evidence here is also caused by an anatomical/functional heterogeneity? Work by

Arnsten and colleagues has shown that particularly the more anterior part of the PrL is involved in working memory and receives most noradrenergic innervation.

[Response] We thank the reviewer for leading us to consider alternative explanations. Indeed, our findings point to the possibility of functional heterogeneity in rodent PrL/mPFC.

We now note in the *Supplementary Information: Discussion* section (line 566~586): “There is functional and anatomical evidence that rodents do not have a dorsolateral PFC analogous to primates ²⁹; instead, many dorsolateral PFC functions, such as working memory, are features of the primordial rodent mPFC/PrL ³⁰. The rat PrL spans from approximately 2.5 mm to 5 mm AP relative to Bregma ³¹. While most studies of rodent PrL in working memory have focused on the PrL AP mid-point ³²⁻³⁵, also used for the PrL ROI here (~3.2 mm AP), there is some evidence that working memory is most susceptible to noradrenergic ³³ and dopaminergic ³⁶ manipulations of the anterior and posterior PrL, respectively, hinting at an anatomical gradient in PrL working memory function. However, other studies have demonstrated that neurons across the entirety of rat PrL encode working memory ³⁷ and that pyramidal neurons at the posterior-most coordinates of PrL in mouse are also critical for working memory ³⁸. Given this mixed evidence, it is possible that the conflicting evidence here is also caused by an anatomical/functional heterogeneity as suggested by the reviewer. Critically, in the context of the present study, Lu and colleagues ³⁹ suggest that the entire medial ridge (including the AP extent of PrL) is involved in the rat DMN, while Tsai and colleagues ⁴⁰ show the AP extent of PrL involved in the rat SN. Taken together, dynamic functional heterogeneity, as well as anatomical and neurochemical along the AP axis of PrL, may contribute to its variable association with the DMN and SN. More precise evidence for integration rather than segregation of DMN and SN within the rodent mPFC, supported by distinct mPFC neuron populations, may be obtained by future studies employing spatially resolved recordings of neuronal activity.”

We point readers to this additional discussion on PrL/mPFC heterogeneity on lines 391~393 of the main text.

Reviewer #3

[Overall Comment] The manuscript is of substantial interest, as it concerns causal manipulations triggering brain state changes related to the DMN. As such, it is not the first to use this approach, and previous work in this area (PNAS 115 (6) 1352-1357, SCI REP; 9: 2570. and others) should be mentioned and discussed in relation to the present findings. This above mentioned work concerns long range corticopetal projections that activate or deactivate the DMN, rather than interfering directly in the cortex. The previous work mostly focused on behavioral and electrophysiological changes, so the present study is to my knowledge one of the first to combine a causal approach with whole brain fMRI, which is highly commendable, but other work exists here too, eg iScience 23(9):101455. It would be useful to add a section in the discussion as to what degree the present findings are consistent with these previous studies. On a general level, the work appears to be well executed, although at times I feel it could be a little

more coherently organized (see also below). The scientific bottom line of subs-specialisation of PRL in the rat is interesting, as is the relation to the cingulate. Reading the manuscript, I was wondering if some of these findings might be related to neuroanatomical data on area connection strength for example (if such work exists in the literature regarding the AI cortex (where are its major outputs going to). It would be of interest to see a short section in the discussion in this regard. The points below occurred to me as I was reading the manuscript, I realize that some of them are addressed actually later, but maybe the authors find this information useful for streamlining the manuscript.

[Response] We appreciate the reviewer's feedback that our manuscript is of substantial interest and for bringing several important pieces of the relevant literature to our attention.

We have added the following additional paragraph relating our findings to the literature on deep-brain structures in DMN function and regulation to the *Supplementary Information: Discussion* section (line 638~663): *“Recent studies have implicated deep-brain structures with long-range, widespread projections in DMN function and regulation (see Aguilar et al.,⁴¹ for a comprehensive review). For example, in rats, Nair et al., 2018 found that spontaneous, basal forebrain (BF), gamma-band activity was elevated during DMN-related behaviors and suppressed during other activities⁴². Subsequent rodent studies employing targeted BF stimulation or inhibition have shown that this structure can regulate the DMN. As such, nonspecific BF stimulation and stimulation of GABAergic BF projections at gamma frequencies enhances DMN-related behavioral and electrophysiological phenotypes and suppresses SN-related behaviors^{43,44}, whereas BF inhibition has the opposite effects⁴³, and inhibition of GABAergic basal BF neurons can reduce ketamine-induced, DMN-related, cortical gamma band power⁴⁵. Tonic stimulation of GABAergic BF neurons⁴⁵ or suspected disinhibition of the BF by inhibition of somatostatin-positive BF neurons^{46,47} produces a complex phenotype suggestive of a loss of DMN regulation and cortical hyper-excitability. While the aforementioned studies focused on specific anterior DMN nodes and point to the involvement of BF GABAergic transmission, Peeters et. al., 2020 found a global reduction in DMN functional-connectivity as a result of BF cholinergic stimulation during fMRI⁴⁸, indicating a potentially cholinergic-specific role of the BF in DMN modulation. In addition, we have previously shown that tonic activation of locus coeruleus (LC) adrenergic projections decreases activity but increases functional connectivity within the anterior DMN and strengthens anticorrelated coupling between the anterior DMN and the AI⁴⁹ — thus, this system could also promote DMN-SN network switching. Notably, both the BF⁵⁰ and LC⁵¹ are reciprocally connected with the AI; therefore, although the specific functions of these connections have yet to be investigated, it is plausible that deep-brain regulation of the DMN includes engagement and/or modulation of the AI-centered effects and SN-DMN interactions elucidated in the present study.”*

Also, reference to this new supplemental discussion can be found in the main manuscript on line 396~400.

Further, as presented in our responses to comments #2.4 and #2.6, we now also provide detailed discussion on the subspecialization of rodent mPFC structures and how our findings relate to the neuroanatomical data regarding the AI cortex excitatory projections.

[Comment #3.1] Line.178: The authors document expression of Chronos-YFP and YFP in the anterior insula, but I did not see any details about which cell types these were expressed in. Histological analyses would be useful here or pointing to data from publications that have employed the same viral vector and described targeted cell types.

[Response] We thank the reviewer for their comment. As detailed in our response to comment #2.1, in the *Supplementary Information: Result* section we now include evidence from the literature for the expected cell-type expression profile of Chronos and EYFP in this study (line 165~185).

[Comment #3.2] Line.199: The description here along with the fig.2 does not appear to detail any findings but explains the procedure. This might be stated explicitly, or this information moved to supplement.

[Response] We thank the reviewer for this suggestion. We believe that **Figure 2** could help the readers to follow the overall analysis strategy and procedures used in our study. Following the reviewer's suggestion, we have included the following sentence in the revised manuscript (line 187): "*Our overall analysis strategy and procedures using BSDS are illustrated in Figure 2.*"

[Comment #3.3] Line.208: According to what criteria were 5 brain states defined? I expect that there is also probably some divergence among the results of the individual rats. Please elaborate. It should be mentioned that other states will be discussed later.

[Response] We apologize for the lack of clarity. BSDS incorporates a learning algorithm which was implemented at the group-level first, with the advantage that the ensuing states are matched across individuals and therefore can be compared in a common framework. It is also possible to apply the BSDS learning algorithm at the individual level but then the states are mismatched and their dynamic properties cannot be directly compared. Specifically, BSDS first takes sequences of ROI time-series from all the rats as input data and infers group-level model parameters, including number of brain states. Then, BSDS uses the group-level learned model parameters as the input for subject-level analysis and estimates the rat-specific temporal evolution of states. Consequently, while the number of states is shared among the rats, the temporal evolution of the brain states over time may varies across individual rats. Nevertheless, please note that for all the analyses performed in our study, such as comparison of occupancy rate of ON and OFF states, we tested and confirmed statistical significance across individual rats.

To determine the optimal number of brain states as well as other model parameters, we used a variational Bayesian (VB) framework⁵², which is a widely used inference method in machine learning. Under the VB framework, BSDS treats the number of states as a random variable and initialize it to a relatively large number. The optimal number of states is learned automatically from the data, using an automatic relevance determination approach in the VB framework, which prunes away a large number of irrelevant states not supported by the data while a few states survive that are supported by the data⁵³. In this study, BSDS models were initialized with 10 brain states for both

Chronos and EYFP rats. Consequently, BSDS identified 5 brain states in the Chronos rats and 3 brain states in the EYFP rats, respectively.

These details have been added to the *Methods* section in the revised manuscript (line 682~687).

[Comment #3.4] Line.224: There do not seem to be any error bars on the 50% SVM classification estimate.

[Response] We appreciate the reviewer's feedback. We have now included error bars on the 50% SVM classification estimate for the EYFP controls in **Figure 3C** of the revised manuscript (**Figure R8**). As hypothesized, the EYFP controls did not distinguish the stimulation and control conditions.

Figure R8: Classification analysis based on a linear Support Vector Machine classifier. Data are represented as mean and standard error of the mean. ** $p < 0.001$, two-tailed t-test, FDR-corrected.

[Comment #3.5] Line.252: The other 3 states are not mentioned further, but authors focus only on states 1 and 2. Coming back to my point above, why were 5 states chosen as cutoff in the first place, and are results identical if 2 states are chosen. In general, have the authors analyzed the data using more standard fMRI methods, such as for example GLM? What are the results of such an analysis; these should be able to capture the same effects as reported here.

[Response] As we described above in our response to **Comment #3.3**, the number of states was not determined manually, but rather using automatic relevance determination procedures implemented in a Variational Bayesian framework⁵² that infer the optimal number of states to best describe the data. This procedure uncovered 5 states for the Chronos group. We did not preset the number of states to 2 because the goal of our study was to investigate dynamic spatiotemporal properties associated with AI stimulation, including transient and short-lived states that are not time-locked to the stimulation. Using this approach, we have shown that it is possible to detect transition states whose engagement uniquely predicts behavior⁵³. Such states and their dynamic properties cannot be identified using GLM as their onset and offset are not a priori, and instead have to be inferred through model-based analysis.

We have indeed analyzed these data using GLM, with explanations, results, and discussion available in the *Supplementary Information: Materials and Methods*, line 119~137; *Results*, line 188~202; *Discussion*, line 534~552; **Figure S2**.

Briefly, GLM did not capture the same effects that we reported using BSDS. The lack of sensitivity of conventional approaches, including GLM, is likely related to an inability to capture the complex profile of temporal response patterns elicited by the stimulation. BSDS belongs to a class of latent space switching models^{54,55} which goes beyond conventional methods, and has been shown to be highly effective in recovering the structure of non-stationary time-varying organization of neural circuits^{53,56,57}. Our computational model-based approach, BSDS⁵³, has the following key advantages over other approaches (*Supplementary Information: Discussion* section, line 475~507):

1. First, BSDS does not require arbitrary sliding windows nor does it impose temporal boundaries associated with predefined task conditions - this is contrast to previous approaches for characterizing dynamic interactions in the brain that have primarily been based on sliding window or clustering techniques (e.g., ICA or PCA-based approaches) applied to observed fMRI data⁵⁸⁻⁶⁰. These previous methods rely on ad hoc procedures for determining critical parameters, such as the window length and number of brain states (i.e., clusters), which are known to greatly influence the estimation of dynamic brain states and connectivity⁶¹. In contrast, BSDS uses a Bayesian framework to automatically regulate model complexity and directly estimates the optimal number of latent states.
2. Second, BSDS applies a hidden Markov model (HMM) to latent space variables of the observed fMRI data, resulting in a parsimonious model of generators underlying the observed data – whereas previous approaches have applied hidden Markov models directly to observed fMRI data without estimating the underlying latent states⁶²⁻⁶⁵. Instead, BSDS applies HMM to latent space variables generated by an autoregressive process, resulting in greater robustness to abrupt and noisy local changes in the observed data and more robust state identification.
3. Third, BSDS allows us to uncover ‘latent’ brain states and their dynamic spatiotemporal properties, including probability and sequence of state transitions as well as inter-regional functional connectivity associated with each brain state, in an optimal latent subspace. In our previous study (Taghia et al⁵³), we demonstrated that BSDS has greater robustness to the abrupt noisy and local changes in fMRI, resulting in more accurately identifying brain states and their temporal dynamic properties than conventional data driven approaches (e.g., ICA or PCA-based approaches). Based on these advantages, we used BSDS in our study to investigate latent brain state dynamics underlying optogenetic manipulation of the AI.

Taken together, our analyses demonstrate that BSDS can capture dynamic circuit mechanisms of optogenetic stimulation that are missed by conventional analyses.

[Comment #3.6] Line.395: The authors observe PRL activation in some segments in ON blocks, as well as increased PRL-CG coupling on ON blocks but no activation of CG on ON blocks. How is this possible? This is an interesting finding; it is worth discussing in a little detail how these 3 effects can occur at the same time. This discussion is actually presented in L.450, it would seem that streamlining the discussion might be helpful to the reader.

[Response] We thank the reviewer for this insightful question. While it is generally assumed that the increased functional connectivity between two regions requires a concurrent increase or decrease in activity in both regions, it is not a necessary condition. An instantaneous temporal pattern of a region could vary without a change in average power, resulting in changing functional connectivity (i.e., correlation) with other regions. For instance, Krieger-Redwood et al. ⁶⁶, reported increased coupling of posterior cingulate cortex (pCC) with prefrontal regions despite pCC deactivation and prefrontal activation during demanding tasks, demonstrating a heterogeneous relationship between activation profiles and functional coupling. Indeed, in a recent study characterizing the rodent SN, Cg and PrL were implicated as nodes of the SN in addition to their putative roles in the DMN ⁴⁰. We believe our results demonstrate functional heterogeneity of these two regions, which is an important research question to be addressed in more in detail in future studies. Following the reviewer's suggestion, we have included this note in the *Supplementary Information: Discussion* section (line 616~631), and refer readers to this information on line 451~453 of the main manuscript.

[Comment #3.7] Line.495: The material related to the transition states seems speculative and does not provide any major advances in understanding; focusing the manuscript on the major discoveries might be advantageous.

[Response] We appreciate the reviewer's thoughtful suggestion. We believe that model-based analyses advance understanding of the spatiotemporal dynamic process underlying SN-DMN interactions. Importantly, transition states are difficult to identify without unsupervised probabilistic learning models, such as those implemented in BSDS, which provide a mechanism for capturing latent nonlinear changes in brain connectivity. Our results demonstrate existence of the transition state that connects brain states related to stimulation ON and OFF blocks. (Please see line 299~323 of the *Supplementary Information: Results* section for more details). This finding is important because it suggests that previous studies, which have almost always focused on predefined task/block boundaries to examine functional connectivity, are likely to miss key features of brain state dynamics and the unique functional circuits associated with them.

We have added emphasis for this point to the Discussion section of the main manuscript on line 516~519:

“Taken together, these findings suggest that previous studies, which have almost always focused on predefined task/block boundaries to examine functional connectivity, are likely to miss key features of brain state dynamics and the unique functional circuits associated with them.”

References

- 1 Chen, S., Langley, J., Chen, X. & Hu, X. Spatiotemporal modeling of brain dynamics using resting-state functional magnetic resonance imaging with Gaussian hidden Markov model. *Brain connectivity* **6**, 326-334 (2016).
- 2 Mandino, F. *et al.* Animal Functional Magnetic Resonance Imaging: Trends and Path Toward Standardization. *Front Neuroinform* **13**, 78 (2019). <https://doi.org/10.3389/fninf.2019.00078>
- 3 Pais-Roldán, P. *et al.* Contribution of animal models toward understanding resting state functional connectivity. *Neuroimage* **245**, 118630 (2021). <https://doi.org/10.1016/j.neuroimage.2021.118630>
- 4 Fukuda, M., Vazquez, A. L., Zong, X. & Kim, S.-G. Effects of the α 2-adrenergic receptor agonist dexmedetomidine on neural, vascular and BOLD fMRI responses in the somatosensory cortex. *European Journal of Neuroscience* **37**, 80-95 (2013). [https://doi.org:https://doi.org/10.1111/ejn.12024](https://doi.org/https://doi.org/10.1111/ejn.12024)
- 5 Lee, S. H. *et al.* An isotropic EPI database and analytical pipelines for rat brain resting-state fMRI. *Neuroimage* **243**, 118541 (2021). <https://doi.org/10.1016/j.neuroimage.2021.118541>
- 6 Paasonen, J., Stenroos, P., Salo, R. A., Kiviniemi, V. & Gröhn, O. Functional connectivity under six anesthesia protocols and the awake condition in rat brain. *NeuroImage* **172**, 9-20 (2018). <https://doi.org:https://doi.org/10.1016/j.neuroimage.2018.01.014>
- 7 Reimann, H. M. & Niendorf, T. The (Un)Conscious Mouse as a Model for Human Brain Functions: Key Principles of Anesthesia and Their Impact on Translational Neuroimaging. *Front Syst Neurosci* **14**, 8 (2020). <https://doi.org/10.3389/fnsys.2020.00008>
- 8 Gutierrez-Barragan, D. *et al.* Unique spatiotemporal fMRI dynamics in the awake mouse brain. *Current Biology* **32**, 631-644. e636 (2022).
- 9 Liang, Z., Liu, X. & Zhang, N. Dynamic resting state functional connectivity in awake and anesthetized rodents. *Neuroimage* **104**, 89-99 (2015). <https://doi.org/10.1016/j.neuroimage.2014.10.013>
- 10 Nieuwenhuis, B. *et al.* Optimization of adeno-associated viral vector-mediated transduction of the corticospinal tract: comparison of four promoters. *Gene therapy* **28**, 56-74 (2021).
- 11 Gehrlach, D. A. *et al.* A whole-brain connectivity map of mouse insular cortex. *Elife* **9**, e55585 (2020).
- 12 Watakabe, A. *et al.* Comparative analyses of adeno-associated viral vector serotypes 1, 2, 5, 8 and 9 in marmoset, mouse and macaque cerebral cortex. *Neuroscience Research* **93**, 144-157 (2015). <https://doi.org:https://doi.org/10.1016/j.neures.2014.09.002>
- 13 Nathanson, J. L., Yanagawa, Y., Obata, K. & Callaway, E. M. Preferential labeling of inhibitory and excitatory cortical neurons by endogenous tropism of adeno-associated virus and lentivirus vectors. *Neuroscience* **161**, 441-450 (2009). <https://doi.org/10.1016/j.neuroscience.2009.03.032>
- 14 Anikeeva, P. *et al.* Optetrode: a multichannel readout for optogenetic control in freely moving mice. *Nat Neurosci* **15**, 163-170 (2011). <https://doi.org/10.1038/nn.2992>
- 15 Gogolla, N. The insular cortex. *Current Biology* **27**, R580-R586 (2017). <https://doi.org:https://doi.org/10.1016/j.cub.2017.05.010>

- 16 Sridharan, D., Levitin, D. J. & Menon, V. A critical role for the right fronto-insular cortex in switching between central-executive and default-mode networks. *Proceedings of the National Academy of Sciences* **105**, 12569-12574 (2008).
- 17 Cai, W., Ryali, S., Pasumarthy, R., Talasila, V. & Menon, V. Dynamic causal brain circuits during working memory and their functional controllability. *Nature communications* **12**, 1-16 (2021).
- 18 Cottam, W. J., Iwabuchi, S. J., Drabek, M. M., Reckziegel, D. & Auer, D. P. Altered connectivity of the right anterior insula drives the pain connectome changes in chronic knee osteoarthritis. *Pain* **159**, 929-938 (2018). <https://doi.org:10.1097/j.pain.0000000000001209>
- 19 Goulden, N. *et al.* The salience network is responsible for switching between the default mode network and the central executive network: replication from DCM. *Neuroimage* **99**, 180-190 (2014). <https://doi.org:10.1016/j.neuroimage.2014.05.052>
- 20 Menon, V. & Uddin, L. Q. Saliency, switching, attention and control: a network model of insula function. *Brain Struct Funct* **214**, 655-667 (2010). <https://doi.org:10.1007/s00429-010-0262-0>
- 21 Sridharan, D., Levitin, D. J. & Menon, V. A critical role for the right fronto-insular cortex in switching between central-executive and default-mode networks. *Proc Natl Acad Sci U S A* **105**, 12569-12574 (2008). <https://doi.org:10.1073/pnas.0800005105>
- 22 Qadir, H. *et al.* Structural connectivity of the anterior cingulate cortex, claustrum, and the anterior insula of the mouse. *Frontiers in neuroanatomy* **12**, 100 (2018).
- 23 Stacho, M. & Manahan-Vaughan, D. Mechanistic flexibility of the retrosplenial cortex enables its contribution to spatial cognition. *Trends in Neurosciences* (2022).
- 24 Zingg, B. *et al.* Neural networks of the mouse neocortex. *Cell* **156**, 1096-1111 (2014).
- 25 van Groen, T. & Michael Wyss, J. Connections of the retrosplenial granular a cortex in the rat. *Journal of Comparative Neurology* **300**, 593-606 (1990). <https://doi.org:https://doi.org/10.1002/cne.903000412>
- 26 van Groen, T. & Wyss, J. M. Connections of the retrosplenial dysgranular cortex in the rat. *Journal of Comparative Neurology* **315**, 200-216 (1992). <https://doi.org:https://doi.org/10.1002/cne.903150207>
- 27 Van Groen, T. & Wyss, J. M. Connections of the retrosplenial granular b cortex in the rat. *Journal of Comparative Neurology* **463**, 249-263 (2003). <https://doi.org:https://doi.org/10.1002/cne.10757>
- 28 Chrastil, E. R. Heterogeneity in human retrosplenial cortex: A review of function and connectivity. *Behav Neurosci* **132**, 317-338 (2018). <https://doi.org:10.1037/bne0000261>
- 29 Seamans, J. K., Lapish, C. C. & Durstewitz, D. Comparing the prefrontal cortex of rats and primates: insights from electrophysiology. *Neurotox Res* **14**, 249-262 (2008). <https://doi.org:10.1007/bf03033814>
- 30 Arnsten, A. F. T. Catecholamine modulation of prefrontal cortical cognitive function. *Trends in Cognitive Sciences* **2**, 436-447 (1998). [https://doi.org:https://doi.org/10.1016/S1364-6613\(98\)01240-6](https://doi.org:https://doi.org/10.1016/S1364-6613(98)01240-6)
- 31 Paxinos, G. & Watson, C. The rat brain in stereotaxic coordinates sixth edition by. *Acad Press* **170**, 10.1016 (2006).
- 32 Datta, D. *et al.* Glutamate Carboxypeptidase II in Aging Rat Prefrontal Cortex Impairs Working Memory Performance. *Front Aging Neurosci* **13**, 760270 (2021). <https://doi.org:10.3389/fnagi.2021.760270>

- 33 Ramos, B. P. *et al.* The Beta-1 Adrenergic Antagonist, Betaxolol, Improves Working Memory Performance in Rats and Monkeys. *Biological Psychiatry* **58**, 894-900 (2005).
[https://doi.org:https://doi.org/10.1016/j.biopsych.2005.05.022](https://doi.org/10.1016/j.biopsych.2005.05.022)
- 34 Wang, M. *et al.* α 2A-Adrenoceptors Strengthen Working Memory Networks by Inhibiting cAMP-HCN Channel Signaling in Prefrontal Cortex. *Cell* **129**, 397-410 (2007).
[https://doi.org:https://doi.org/10.1016/j.cell.2007.03.015](https://doi.org/10.1016/j.cell.2007.03.015)
- 35 Yang, S.-T., Shi, Y., Wang, Q., Peng, J.-Y. & Li, B.-M. Neuronal representation of working memory in the medial prefrontal cortex of rats. *Molecular Brain* **7**, 61 (2014). [https://doi.org:10.1186/s13041-014-0061-2](https://doi.org/10.1186/s13041-014-0061-2)
- 36 Zahrt, J., Taylor, J. R., Mathew, R. G. & Arnsten, A. F. T. Supranormal Stimulation of D₁ Dopamine Receptors in the Rodent Prefrontal Cortex Impairs Spatial Working Memory Performance. *The Journal of Neuroscience* **17**, 8528-8535 (1997). [https://doi.org:10.1523/jneurosci.17-21-08528.1997](https://doi.org/10.1523/jneurosci.17-21-08528.1997)
- 37 Yang, S. T., Shi, Y., Wang, Q., Peng, J. Y. & Li, B. M. Neuronal representation of working memory in the medial prefrontal cortex of rats. *Mol Brain* **7**, 61 (2014). [https://doi.org:10.1186/s13041-014-0061-2](https://doi.org/10.1186/s13041-014-0061-2)
- 38 Vogel, P., Hahn, J., Duvarci, S. & Sigurdsson, T. Prefrontal pyramidal neurons are critical for all phases of working memory. *Cell Reports* **39**, 110659 (2022).
- 39 Lu, H. *et al.* Rat brains also have a default mode network. *Proceedings of the National Academy of Sciences* **109**, 3979-3984 (2012).
- 40 Tsai, P.-J. *et al.* Converging structural and functional evidence for a rat salience network. *Biological Psychiatry* **88**, 867-878 (2020).
- 41 Aguilar, D. D. & McNally, J. M. Subcortical control of the default mode network: Role of the basal forebrain and implications for neuropsychiatric disorders. *Brain Research Bulletin* **185**, 129-139 (2022).
[https://doi.org:https://doi.org/10.1016/j.brainresbull.2022.05.005](https://doi.org/10.1016/j.brainresbull.2022.05.005)
- 42 Nair, J. *et al.* Basal forebrain contributes to default mode network regulation. *Proceedings of the National Academy of Sciences* **115**, 1352-1357 (2018). [https://doi.org:doi:10.1073/pnas.1712431115](https://doi.org/10.1073/pnas.1712431115)
- 43 Klaassen, A.-L., Heiniger, A., Vaca Sánchez, P., Harvey, M. A. & Rainer, G. Ventral pallidum regulates the default mode network, controlling transitions between internally and externally guided behavior. *Proceedings of the National Academy of Sciences* **118**, e2103642118 (2021).
[https://doi.org:doi:10.1073/pnas.2103642118](https://doi.org/10.1073/pnas.2103642118)
- 44 Lozano-Montes, L. *et al.* Optogenetic Stimulation of Basal Forebrain Parvalbumin Neurons Activates the Default Mode Network and Associated Behaviors. *Cell Reports* **33**, 108359 (2020).
[https://doi.org:https://doi.org/10.1016/j.celrep.2020.108359](https://doi.org/10.1016/j.celrep.2020.108359)
- 45 McNally, J. M. *et al.* Optogenetic manipulation of an ascending arousal system tunes cortical broadband gamma power and reveals functional deficits relevant to schizophrenia. *Molecular Psychiatry* **26**, 3461-3475 (2021). [https://doi.org:10.1038/s41380-020-0840-3](https://doi.org/10.1038/s41380-020-0840-3)
- 46 Espinosa, N., Alonso, A., Lara-Vasquez, A. & Fuentealba, P. Basal forebrain somatostatin cells differentially regulate local gamma oscillations and functionally segregate motor and cognitive circuits. *Scientific Reports* **9**, 2570 (2019). [https://doi.org:10.1038/s41598-019-39203-4](https://doi.org/10.1038/s41598-019-39203-4)

- 47 Espinosa, N. *et al.* Basal Forebrain Gating by Somatostatin Neurons Drives Prefrontal Cortical Activity. *Cerebral Cortex* **29**, 42-53 (2017). <https://doi.org:10.1093/cercor/bhx302>
- 48 Peeters, L. M. *et al.* Cholinergic Modulation of the Default Mode Like Network in Rats. *iScience* **23**, 101455 (2020). <https://doi.org:https://doi.org/10.1016/j.isci.2020.101455>
- 49 Oyarzabal, E. A. *et al.* Chemogenetic stimulation of tonic locus coeruleus activity strengthens the default mode network. *Sci Adv* **8**, eabm9898 (2022). <https://doi.org:10.1126/sciadv.abm9898>
- 50 Do, J. P. *et al.* Cell type-specific long-range connections of basal forebrain circuit. *eLife* **5**, e13214 (2016). <https://doi.org:10.7554/eLife.13214>
- 51 Gompf, H. S. *et al.* Locus ceruleus and anterior cingulate cortex sustain wakefulness in a novel environment. *J Neurosci* **30**, 14543-14551 (2010). <https://doi.org:10.1523/jneurosci.3037-10.2010>
- 52 Jordan, M. I., Ghahramani, Z., Jaakkola, T. S. & Saul, L. K. An Introduction to Variational Methods for Graphical Models. *Machine Learning* **37**, 183-233 (1999). <https://doi.org:10.1023/A:1007665907178>
- 53 Taghia, J. *et al.* Uncovering hidden brain state dynamics that regulate performance and decision-making during cognition. *Nature communications* **9**, 1-19 (2018).
- 54 Semedo, J., Zandvakili, A., Kohn, A., Machens, C. K. & Byron, M. Y. in *Advances in neural information processing systems*. 2942-2950.
- 55 Glaser, J. I., Whiteway, M. R., Cunningham, J. P., Paninski, L. & Linderman, S. W. Recurrent switching dynamical systems models for multiple interacting neural populations. *bioRxiv* (2020).
- 56 Lee, B. *et al.* Latent brain state dynamics and cognitive flexibility in older adults. *Progress in Neurobiology*, 102180 (2021).
- 57 Cai, W. *et al.* Latent brain state dynamics distinguish behavioral variability, impaired decision-making, and inattention. *Molecular Psychiatry*, 1-14 (2021).
- 58 Cai, W., Chen, T., Szegletes, L., Supekar, K. & Menon, V. Aberrant time-varying cross-network interactions in children with attention-deficit/hyperactivity disorder and the relation to attention deficits. *Biological Psychiatry: Cognitive Neuroscience and Neuroimaging* **3**, 263-273 (2018).
- 59 Rashid, B. *et al.* Classification of schizophrenia and bipolar patients using static and dynamic resting-state fMRI brain connectivity. *Neuroimage* **134**, 645-657 (2016).
- 60 Allen, E. A. *et al.* Tracking whole-brain connectivity dynamics in the resting state. *Cerebral cortex* **24**, 663-676 (2014).
- 61 Hutchison, R. M. *et al.* Dynamic functional connectivity: promise, issues, and interpretations. *Neuroimage* **80**, 360-378 (2013).
- 62 Taghia, J. *et al.* Bayesian switching factor analysis for estimating time-varying functional connectivity in fMRI. *Neuroimage* **155**, 271-290 (2017). <https://doi.org:10.1016/j.neuroimage.2017.02.083>
- 63 Vidaurre, D., Smith, S. M. & Woolrich, M. W. Brain network dynamics are hierarchically organized in time. *Proceedings of the National Academy of Sciences* **114**, 12827-12832 (2017).
- 64 Vidaurre, D. *et al.* Spectrally resolved fast transient brain states in electrophysiological data. *Neuroimage* **126**, 81-95 (2016).

- 65 Ryali, S. *et al.* Temporal Dynamics and Developmental Maturation of Salience, Default and Central-Executive Network Interactions Revealed by Variational Bayes Hidden Markov Modeling. *PLoS Comput Biol* **12**, e1005138 (2016). <https://doi.org:10.1371/journal.pcbi.1005138>
- 66 Krieger-Redwood, K. *et al.* Down but not out in posterior cingulate cortex: Deactivation yet functional coupling with prefrontal cortex during demanding semantic cognition. *NeuroImage* **141**, 366-377 (2016). <https://doi.org:https://doi.org/10.1016/j.neuroimage.2016.07.060>

REVIEWER COMMENTS

Reviewer #1 (Remarks to the Author):

The authors have done an excellent job of responding to my comments and the comments of the other reviewers. I appreciate the extra analysis and clarifications, and support publication.
Shella Keilholz

Reviewer #2 (Remarks to the Author):

The authors have adequately addressed my previous concerns.

Re: NCOMMS-22-25913B, Optogenetic stimulation of anterior insular cortex neurons reveals causal mechanisms underlying suppression of the default mode network by the salience network

Reviewer #1

[Overall Comments] The authors have done an excellent job of responding to my comments and the comments of the other reviewers. I appreciate the extra analysis and clarifications, and support publication.

[Response] We thank the reviewer for the positive comments and careful review, which helped improve the manuscript.

Reviewer #2

[Overall Comments] The authors have adequately addressed my previous concerns.

[Response] We thank the reviewer for the constructive and insightful comments, which have helped us to substantially improve our manuscript.